



# Best practices in software development for robust and reproducible geoscientific models based on insights from the Global Carbon Project models

Konstantin Gregor[1], Benjamin F. Meyer[1], Tillmann Gaida[2], Victor Justo Vasquez[3], Karina Bett-Williams[4,5], Matthew Forrest[6], João P. Darela-Filho[1], Sam Rabin[7], Marcos Longo[8], Joe R. Melton[9], Johan Nord[10], Peter Anthoni[11], Vladislad Bastrikov[12], Thomas Colligan[13,14], Christine Delire[15], Michael C. Dietze[16], George Hurtt[17], Akihiko Ito[18], Lasse T. Keetz[19], Jürgen Knauer[20], Johannes Köster[21], Tzu-Shun Lin[7], Lei Ma[17], Marie Minvielle[15], Stefan Olin[10], Sebastian Ostberg[22], Hao Shi[23], Reiner Schnur[24], Urs Schönenberger[25], Qing Sun[26,27,28], Peter E. Thornton[29], and Anja Rammig[1]

[1]TUM School of Life Sciences, Technical University of Munich, Freising, Germany
[2]Goto10 GmbH, Munich, Germany
[3]Thoughtworks GmbH, Hamburg, Germany
[4]Global Systems Institute, University of Exeter, Exeter EX4 4PY, UK
[5]UK Met Office, Fitzroy Road, Exeter EX1 3PB, UK
[6]Seckenberg Biodiversity and Climate Research Centre, Frankfurt, Germany
[7]NSF National Center for Atmospheric Research, Boulder, Colorado, USA
[8]Climate and Ecosystem Sciences Division, Lawrence Berkeley National Laboratory, Berkeley, CA, 94720, USA
[9]Climate Research Division, Environment, and Climate Change Canada, Victoria, BC, V8N 1V8, Canada
[10]Department of Physical Geography and Ecosystem Science, Lund University, Sölvegatan 12, 22362 Lund, Sweden
[11]Karlsruhe Institute of Technology, Institute of Meteorology and Climate, Research/Atmospheric Environmental Research, 82467 Garmisch-Partenkirchen, Germany
[12]Science Partners, Paris 75010, France
[13]University of Maryland, College Park, MD 20742, USA
[14]NASA Goddard Space Flight Center, Earth Sciences Division, Biospheric Sciences Lab, 6 Greenbelt, MD 20771, USA
[15]CNRM, Météo-France, CNRS, Université de Toulouse, Toulouse, France
[16]Department of Earth & Environment, Boston University, Boston, MA 02215, USA
[17]Department of Geographical Sciences, University of Maryland, College Park, MD 20770, USA
[18]The University of Tokyo, Tokyo, Japan
[19]Department of Geosciences, University of Oslo, Norway
[20]Faculty of Science, University of Technology Sydney, Ultimo, NSW 2007, Australia
[21]Bioinformatics and Computational Oncology, Institute for AI in Medicine (IKIM), University Hospital Essen, University of Duisburg-Essen, Essen, Germany
[22]Potsdam Institute for Climate Impact Research (PIK), Member of the Leibniz Association, Potsdam, Germany
[23]State Key Laboratory for Ecological Security of Regions and Cities, Research Center for Eco-Environmental Sciences, Chinese Academy of Sciences, Beijing 100085, China
[24]Max Planck Institute for Meteorology, Hamburg, Germany
[25]TNG Technology Consulting GmbH, Munich, Germany
[26]Climate and Environmental Physics, Physics Institute, University of Bern, Bern, Switzerland
[27]Wyss Academy for Nature, University of Bern, Bern, Switzerland
[28]Oeschger Centre for Climate Change Research, University of Bern, Bern, Switzerland
[29]Environmental Sciences Division, Oak Ridge National Laboratory, Oak Ridge, TN, USA





**Correspondence:** Konstantin Gregor (konstantin.gregor@posteo.de)

**Abstract.** Computational models play an increasingly vital role in scientific research, by numerically simulating processes that cannot be solved analytically. Such models are fundamental in geosciences and offer critical insights into the impacts of global change on the Earth system today and in the future. Beyond their value as research tools, models are also software products and should therefore adhere to certain established software engineering standards. However, scientists are rarely trained as

software developers, which can lead to potential deficiencies in software quality like unreadable, inefficient, or erroneous code. The complexity of these models, coupled with their integration into broader workflows, also often makes reproducing results, evaluating processes, and building upon them highly challenging.

In this paper, we review the current practices within the development processes of the state-of-the-art land surface models used by the Global Carbon Project. By combining the experience of modelers from the respective research groups with the

expertise of professional software engineers, we bridge the gap between software development and scientific modeling to outline key principles and tools for improving software quality in research. We explore four main areas: 1) model testing and validation, 2) scientific, technical, and user documentation, 3) version control, continuous integration, and code review, and 4) the portability and reproducibility of workflows.

Our review of current models reveals that while modeling communities are incorporating many of the suggested practices,

significant room for improvement remains in areas such as automated testing, documentation, and reproducible workflows. For instance, there is limited adoption of automated documentation and testing, and provision of reproducible workflow pipelines remains an exception. This highlights the need to identify and promote essential software engineering practices within the scientific community. Nonetheless, we also discuss numerous examples of practices within the community that can serve as guidelines for other models and could even help streamline processes within the entire community.

We conclude with an open-source example implementation of these principles built around the LPJ-GUESS model, showcasing portable and reproducible data flows, a continuous integration setup, and web-based visualizations. This example may serve as a practical resource for model developers, users, and all scientists engaged in scientific programming./

## 1 Introduction - models and data analyses as software products

Computational models are becoming increasingly important in all fields of science at all scales. From cellular processes (Chau-

viere et al., 2010) to the human brain (Kringelbach and Deco, 2020), emotions to cultural behaviors (Edelmann et al., 2020), plant growth to global vegetation (Snell et al., 2014), or local weather (Krayenhoff et al., 2021) to global climate (Eyring et al., 2016). Scientific model development includes incorporating new physical, chemical, biological or economic processes, improving computational performance, simulating novel scenarios, and refining evaluation, benchmarking, data integration, or uncertainty propagation methods. But models are not just scientific tools — they are also complex software systems. As

such, they should adhere to modern software engineering standards, promoting correctness, reliability, and quality in areas such as usability, maintainability, portability, reusability, extensibility, and performance. However, scientists usually lack for-





mal training in software engineering, which can hinder the development of high-quality software, although this is essential to create robust models. Furthermore, there is little recognition of proper software development in academia and little funding for software development (Merow et al., 2023), also leading to code that does not adhere to high technical standards. Indeed,

climate models were found to require more efforts in becoming "more readable, maintainable, and portable" (Easterbrook, 2010). Notably, some valuable guidance for coding in scientific computing is given by Wilson et al. (2014, 2017), covering topics like data management, collaboration, and writing code.

Scientific models are typically embedded in broader workflows, encompassing data processing and analyses. These additional components often exhibit even lower software quality, exacerbating reproducibility issues. Despite advances such as

FAIR data policies – standing for data that is findable, accessible, interoperable, and reusable (Wilkinson et al., 2016) – and the growing adoption of open source practices (Barton et al., 2022), the availability of code and data alone does not guarantee the reproducibility of complex workflows. While tools are available to enhance portability and reproducibility (Mölder et al., 2021; Landau, 2021a) and some platforms support running and analyzing geoscientific models anywhere (e.g., Fer et al., 2021; Keetz et al., 2023), these remain exceptions.

Here, we share insights from the land surface model (LSM) community regarding current practices in model development, testing, collaboration, and documentation. For this, we conducted a survey among the 20 LSMs participating in the Global Carbon Project (GCP). The GCP (Friedlingstein et al., 2023) is an international research initiative aimed at quantifying the global carbon cycle. Its primary activity is the annual Global Carbon Budget, which offers detailed analyses of the carbon cycle, and their implications for climate change, offering critical insights for policy-making. The LSMs contribute by simulating

the fluxes of carbon between land and atmosphere (and partly also oceans). through mechanistically modeled processes like photosynthesis, tree mortality, or land-use change.

Our survey highlights that many state-of-the-art LSMs implement valuable software standards and technical processes, offering opportunities for mutual learning. However, there is still room for improvement, and the community can benefit from sharing best practices in areas such as automated testing, continuous integration, portability, documentation, and reproducibil-

ity.

To aid model communities in improving their practices, we consolidate insights from software development professionals and model developers of the GCP-models, highlighting key aspects of software engineering and how they apply in model development. While not exhaustive, we provide ideas for improving the quality of scientific work and offer discussion on areas needing more investment. Readers are encouraged to explore the principles, tools, and languages best suited to their needs.

Notably, specific model examples mentioned here are not intended as endorsements or criticisms. Each of the models and surrounding frameworks offer their particular benefits and limitations which allowed us to extract the most important highlights and areas for improvement. Although we focus on LSMs, most of the concepts are applicable for all sorts of scientific modeling and software.

The following sections cover best practices for writing readable, maintainable, and testable code, as well as methods to

improve correctness through testing and validation. We discuss documentation, version control, code review, and continuous integration as tools for improving collaboration and enforcing high code quality. Strategies to create portable and reproducible





workflows are also addressed, emphasizing the importance of reproducibility. We conclude with an example of a unit-tested, documented, version-controlled, portable, and reproducible workflow using the LPJ-GUESS model, along with pre-processing and data analysis, as well as web-based visualizations of results.

## 2  Summary of the current state of coding in the land surface modeling community

To understand the current situation of software development processes in the modeling community, we conducted a survey among the 20 LSMs of the GCP (Table 1). We asked about various practices including version control, automated documentation, benchmarking, testing, and reproducibility frameworks (survey questions in Sect. B1). We found that the majority (13) of the models are written in Fortran, while three are in C, three in C++, and one in both Fortran and C. Most of the models provide their code publicly to at least some extent. Version control is mostly used (17 models) and the majority uses Git (Fig. 4a). One model uses Subversion, one is moving from Subversion to Git, and one uses both simultaneously. All models using version control also use a version control platform like GitHub, but this is not always publicly accessible. For instance, five models use their platform internally, and provide the code via Zenodo or upon request. Main reasons for not going fully public include insufficient compute power or storage capabilities on hosted version control platforms and the preference to keep some collaboration aspects private, also because of a lack of time to address user inquiries. Additionally, there may be a fear of losing a competitive edge if unique features become easily accessible to others. Testing, documentation, and reproducibility practices are diverse across models and will be addressed in later sections.

Before delving into technical aspects, it is essential to understand the current landscape of LSM development and the challenges it faces. Modeling groups vary in size and funding, and many scientists contributing to code development are not professional software developers. Additionally, new researchers with various backgrounds frequently join the field, yet there is often insufficient funding for dedicated scientific programming positions.

Clear guidelines and best practices are crucial to maintain code quality and collaboration. Standards need to be documented to explain contribution processes, coding conventions, review protocols, resource and data management, and bug reporting. This is particularly important for new scientists, who must not only understand the model itself but also navigate the development workflow and learn how to contribute effectively.

Despite the critical role of well-structured code, scientists often have very limited time for software development. Moreover, academic incentives prioritize publications over writing maintainable and robust code and provide even less rewards for maintaining code after its first publication (Merow et al., 2023). However, high-quality code benefits both the community and individual researchers by ensuring correctness, robustness, maintainability, and extensibility. In the following, we explore strategies to address these challenges and improve model development practices.



**Table 1.** The 20 LSMs used in the Global Carbon Budget 2023. The code URL refers to the most up-to-date code available which is not necessarily the one used to run the simulations for the Global Carbon Project.

| Model | References | Language | Public URL (if available) |
|---|---|---|---|
| CABLE-POP | Haverd et al. (2018) | Fortran | https://github.com/CABLE-LSM/CABLE/tree/CABLE-POP_TRENDY |
| CLASSIC | Melton et al. (2020), Seiler et al. (2021) | Fortran | https://gitlab.com/cccma/classic |
| CTSM (CLM) | Lawrence et al. (2019) | Fortran | https://github.com/ESCOMP/CTSM |
| DLEM | Tian et al. (2011, 2015) | C++ | https://chess.auburn.edu/models/ |
| EDv3 | Moorcroft et al. (2001); Ma et al. (2022) | C++ | https://zenodo.org/records/6901510 |
| ELM | Yang et al. (2023); Burrows et al. (2020) | Fortran | https://github.com/E3SM-Project/E3SM |
| ISAM | Jain et al. (2013); Meiyappan et al. (2015); Shu et al. (2020) | Fortran | https://climatemodels.uchicago.edu/isam/isam.doc.html |
| SURFEX/ISBA-CTRIP | Voldoire et al. (2017); Masson et al. (2013); Delire et al. (2020) | Fortran | https://www.umr-cnrm.fr/surfex |
| ICON-Land/JSBACH | Mauritsen et al. (2019); Reick et al. (2021); Schneck et al. (2022) | Fortran | https://icon-model.org |
| JULES-ES | Wiltshire et al. (2021); Sellar et al. (2019); Burton et al. (2019); Best et al. (2011); Clark et al. (2011) | Fortran | https://code.metoffice.gov.uk/trac/jules |
| LPJ-GUESS | Smith et al. (2014); Olin et al. (2015); Lindeskog et al. (2021) | C++ | https://web.nateko.lu.se/lpj-guess/download.html |
| LPJmL | Schaphoff et al. (2018); Von Bloh et al. (2018); Lutz et al. (2019); Heinke et al. (2023) | C | https://github.com/PIK-LPJmL/LPJmL |
| LPJ-EOSIM (LPJ-wsl) | Poulter et al. (2011) | C | https://github.com/LPJ-EOSIM/LPJ-wsl_v2.0 |
| LPX-Bern | Lienert and Joos (2018) | Fortran | https://lpx-bern.github.io/info/about |
| OCN | Zaehle and Friend (2010); Zaehle et al. (2011) | Fortran | |
| ORCHIDEEv3 | Krinner et al. (2005); Vuichard et al. (2019); Zaehle and Friend (2010), Vuichard et al. (2019) | Fortran | https://forge.ipsl.fr/orchidee |
| pIBIS | Yuan et al. (2014) | Fortran, C | https://github.com/jxliu2018/pIBIS |
| SDGVM | Woodward and Lomas (2004); Walker et al. (2017) | Fortran | https://bitbucket.org/walkeranthonyp/sdgvm/src/main |
| VISIT | Ito and Inatomi (2012); Kato et al. (2013) | C | |
| YIBs | Yue and Unger (2015) | Fortran | https://github.com/YIBS01/YIBS_site |



## 3 Code and model correctness: testing and validation

In scientific code, two means of ensuring correctness are crucial: testing (sometimes also called "verification"[1]) and validation. Pipitone and Easterbrook (2012) have defined these two terms as follows: "Validation is the process of checking that the theoretical system properly explains the observational system, and verification is the process of checking that the calculational system correctly implements the theoretical system. [ ... ] 'Are we building the right thing?' (validation) and, 'Are we building the thing right?' (verification [/testing])." Here we give an overview of the methods employed by the GCP models and discuss them.

### 3.1 Testing code correctness with automated tests

Testing code at multiple scales is critical in all fields software engineering. Ideally, every piece of code should be tested to help ensure the correctness of all parts of the model and not just the final output, to avoid equifinality issues ("right answers for the wrong reasons"). With sufficient test coverage, a developer can be confident that their new code does not break existing code, or is given direct feedback when and where the issue occurs, allowing for swift detection of problems. Such a testing infrastructure also allows for a confident and timely integration of new features into the model. Finally, testing can also serve as a means of documentation, as it shows what a function is supposed to do.

We found that only three of the 20 models have an extensive automated testing infrastructure, with 12 models having no automated testing in their code base (Fig. 1). However, many of them have extensive manual testing practices. To facilitate adaptation of this important aspect of software engineering, we provide some insights in the following sections.

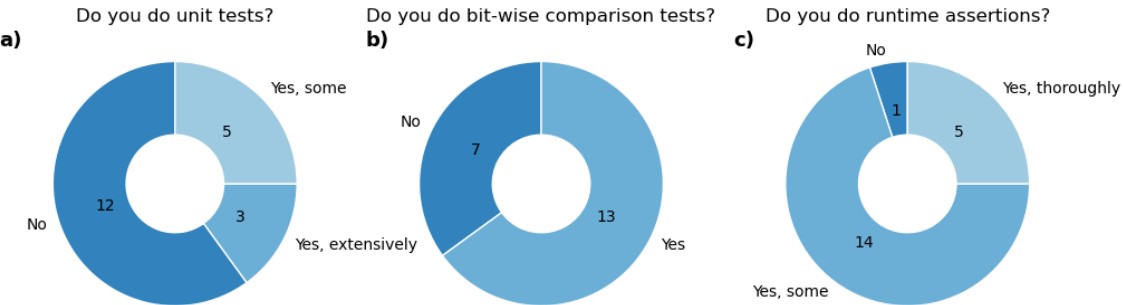

**Figure 1.** Survey results regarding testing infrastructures within the 20 LSMs employed by the GCP. Refer to supplementary files for details on the questions of the survey. The answers depicted are simplifications of the actual answers which were partly also custom answers by the modelers.

---

[1]In software development, "verification" (or "formal verification") of software is the process of proving correct implementation against a formal specification through mathematical means. This is not what is meant here. We therefore refrain from using this term.




### 3.1.1 Writing maintainable and testable code

A first critical step is writing the code such that it is easily understandable, maintainable, and testable. This starts from extracting
code into well-defined and well-named *units* (e.g., functions, subroutines, classes). This avoids duplication and thereby reduces
error-proneness (following the DRY principle, "don't repeat yourself") and allows automated testing of each unit. Numerous
other software paradigms exist that can help make scientific code more maintainable, testable, readable, and understandable
are described in Gregor (2024a, b).

### 3.1.2 The testing pyramid

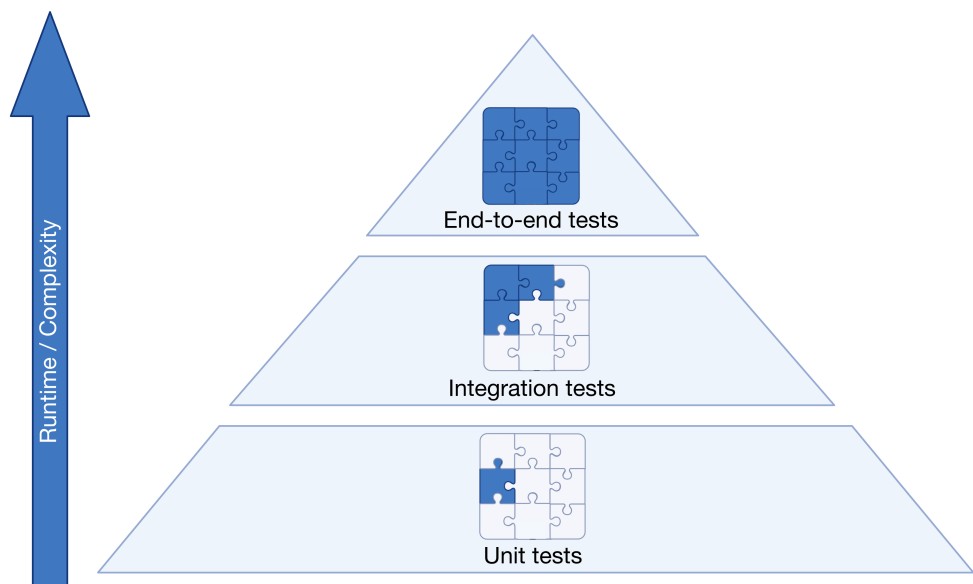

**Figure 2.** Depiction of the testing pyramid as described in Sect. 3.1.2

In professional software development, testing is divided into a pyramid of three layers (Fig. 2). The bottom layer consists of
many small *unit tests*. These should run in milliseconds to seconds such that every developer can run them quickly anytime,
and should test one isolated unit only. Tools like *mocking* frameworks are helpful to replace parts of the code under test with
a "test double" with pre-defined logic (Meszaros, 2009). This allows isolating certain parts of the code for the test and easily
specifying aspects without interfering with other implementations, see code snippet 4 for an example. A benefit of unit tests is
125 that they precisely detect where something is wrong. This means, however, that if a method signature is changed, e.g., by giving
it a new parameter, all tests that call this function need to be adapted as well. Unit tests should follow the FIRST principles
(Martin, 2012): fast, independent, repeatable, self-validating (not requiring user interaction), and timely. *Timely* often means
that testing precedes code writing through *test-driven development (TDD)*, ensuring rigorous testing of all code. However,





this approach may not always suit scientific models, where code is often exploratory and frequently rewritten or discarded. A practical alternative is to first experiment freely to determine the best approach for a new feature. Once the path is clear, one can discard exploratory code and restart using TDD. Since many models lack rigorous testing, an immediate shift to TDD may be overwhelming. Instead, scientific communities could adopt a balanced strategy, writing tests after developing key model components or coherent parts of data analysis. This ensures functionality is validated before building upon the work.

The second layer of testing consists of a number of larger *integration tests*. These test how multiple units work together. For instance, one could simulate one day of the year at a single location, covering multiple processes like phenology, growth, and water uptake, and their impact on radiative and water balance. An advantage of integration tests over unit tests is that they test logical groups of code and do not require extensive refactoring when the underlying code is changed.

At the top of the pyramid are a few large *end-to-end-tests*. These could be full model runs on a regional or global scale and ascertain that the entire model flow is working as expected. These can take a long time to run and as such should only be performed periodically.

Regarding the test strategy for a model, some cost-benefit analysis needs to be done, striking a balance between a solid test base of the model (or model output analysis or data preprocessing pipeline) while not hindering too much the development of new features and model improvements and changes, for instance due to the time it takes to adapt existing tests when parts of the model change. Notably, a "benefit" of scientific code over professional code products is that the latter are usually run on many users' computers. If the problem occurs there, it is already too late. For a scientific model, developer and user are often the same person or they are in close contact, allowing swift correction of bugs. Therefore, test coverage of up to 100%, as often desired in professional software development, might not be necessary. Notably, it might be sensible to focus on integration tests, thereby testing connected logical parts of the model, while fine-grained unit testing is suitable for functions that are highly critical and/or are used in multiple parts of the code. This approach also provides a relatively low-effort path to improving test coverage, especially when it is currently limited. While the testing landscape in the LSM community is quite diverse, a common strategy is to have periodic end-to-end tests, for instance on a nightly or weekly basis.

### 3.1.3 Bit-wise comparison tests

Bit-wise comparisons are common in Earth system models to prevent regressions (Easterbrook and Johns, 2009). New features are controlled via configuration flags, allowing developers to verify that disabling a feature yields identical results bit-for-bit. Indeed, 13 of the GCP models regularly use such tests to ensure new code does not alter existing behavior (Fig. 1b). This requires features to be easily toggled by configuration, ensuring reproducibility of past results with newer model versions. However, unused legacy code should still be removed, and version control systems allow preserving previous implementations. We address bit-wise reproducibility across machines in Sect. 6.2.

### 3.1.4 Assertions

Unit tests are run during development, independent of "production" model runs conducted for scientific studies. Since full model runs might take days or weeks to complete, it is impossible to test all possible cases in a testing suite. Assertions, on the





---

**Code snippet 1** Example of using assertions in data analysis functions, from our example showcase using Python and the Pandas library. Note also the type hint, indicating the type of the variable `lon_lat_dataset`.

---

```python
import pandas as pd

def get_total_value_per_year(lon_lat_dataset: pd.DataFrame, variable_name):
    assert lon_lat_dataset.index.names == ['Lon', 'Lat', 'Year'], "Wrong index"
    assert lon_lat_dataset.index.is_unique, "Index is not unique"
    assert np.all(~np.isnan(lon_lat_dataset)), "NaNs in dataset!"

    areas = lon_lat_dataset.index.get_level_values('Lat').map(get_area_for_lat)
    total_values_per_gridcell = areas * lon_lat_dataset[variable_name]

    return total_values_per_gridcell.groupby('Year').sum()
```

---

other hand, are short checks that test the code at runtime and provide a complementary approach by verifying computations at runtime. They can help ensure that values remain within expected ranges, inputs are valid, and physical laws, such as mass conservation, are upheld.

Beyond validation, assertions naturally document code. Input assertions prevent unexpected inputs, and intermediate assertions show the thought process of the developer. For instance, intermediate checks for matrix dimensions or ensuring values remain positive help make the code readable and understandable.

Notably, assertions in programming languages were initially invented for debugging, not for production code. However, in scientific modeling, critical assertions such as mass balance checks or input validation remain useful even in production.

Performance of the assertions needs to be considered as they may slow down the model. Therefore, it might be useful to distinguish between essential checks like conservation of mass and simple value checks from those that can be omitted for efficiency. Our survey showed that nearly all GCP LSMs already use assertions to some extent, but more progress can be made here (Fig. 1c).

Assertions are particularly valuable in data processing workflows. Data preprocessing scripts will likely only be run once for

one dataset and then never again. The data with which the code has to work is therefore usually known in advance. Thus, tests for the scripts might be obsolete: "Should we invent a test dataset that contains [missing data] we know we don't have for the sake of [testing for] it? That would force us to write new code to address a problem we don't have" (McBain, 2023). Instead, assertions within the processing pipelines help catch unexpected issues early (Snippet 1), pinpointing errors efficiently.

Notably, redundant efforts in writing preprocessing scripts for the same input data is a key issue within the community.

Therefore, the focus needs to lie on tools that address common issues like standardizing input formats, thereby reducing duplication (see Sect. 6.1.3).



## 3.2 Automated testing frameworks

Regarding the tools used for automated testing in the LSM-community, there are numerous ones. While many models have created their own testing frameworks, other models rely on various existing frameworks. These are, namely, Google tests for C++ code, pFUnit[2] for Fortran, Unity[3] for testing of C code, and the tools unittest[4] and pytest[5] for Python code. Furthermore, apart from the well known netCDF tools cdo and nco, nccmp[6] and cprnc[7] are employed in testing pipelines to compare netCDF files.

The LSM community uses a variety of tools for automated testing. While some models have created their own custom testing frameworks, others rely on established tools such as Google Test for C++, pFUnit[8] for Fortran, Unity[9] for C, and unittest, pytest, and coverage for Python. Additionally, beyond the well-known netCDF tools CDO[10] and NCO[11], utilities like nccmp[12] and cprnc[13] are used to compare netCDF files in testing pipelines.

## 3.3 Model validation

Model validation for scientific correctness and evaluation against observations, is a central challenge in scientific model development. Its nature varies depending on discipline, model, research question, studied process, scale, quality, and availability of validation data. It is an inevitable part of the scientific process and often involves manual steps; for instance, comparing global maps of a variable of interest from multiple model runs (Easterbrook and Johns, 2009). However, some parts of this can be automated and efforts can be made such that the tooling makes such investigations as easy as possible.

Whenever possible, benchmarking the model with other datasets should be part of an automated process. Small benchmarks (e.g., runs for a single grid cell) could be part of the automated testing pipeline, checking for instance that the modeled carbon content in vegetation does not deviate too much from observations. For larger benchmarking suites, test runs could be run on a schedule, for instance once per week or before a new release, to keep the computational load manageable. Here it needs to be understood whether automatic tests suffice or whether it is necessary that scientists look at the comparison of model outputs and other datasets to understand whether model performance is still acceptable after changing model code.

The numerous fields of geosciences offer a variety of benchmarking and validation tools. The most prominent examples are the tools designed to assess the Earth system models of CMIP, e.g., ESMValTools (Eyring et al., 2020) and PMP (Lee et al., 2024), which are used to evaluate the relative performance of the models compared to observations. The LSM community

---

[2]https://github.com/Goddard-Fortran-Ecosystem/pFUnit
[3]https://github.com/ThrowTheSwitch/Unity
[4]https://docs.python.org/3/library/unittest.html
[5]http://pytest.org
[6]https://gitlab.com/remikz/nccmp
[7]https://github.com/ESMCI/cprnc
[8]https://github.com/Goddard-Fortran-Ecosystem/pFUnit
[9]https://github.com/ThrowTheSwitch/Unity
[10]https://code.mpimet.mpg.de/projects/cdo
[11]https://nco.sourceforge.net/
[12]https://gitlab.com/remikz/nccmp
[13]https://github.com/ESMCI/cprnc





uses various tools for model validation, the most commonly used is the International Land Model Benchmarking (ILAMB) system which we describe below (Collier et al., 2018a). For hydrometeorological values, the Land surface Verification Tool offers further capabilities for model-data validation (Kumar et al., 2012). We give some further examples from the community
in Sect. A1.

### 3.3.1  ILAMB system

The ILAMB system is a FLOSS ("free, libre, open-source software") package created in the scope of the ILAMB project to support the standardization of land model benchmarking (Hoffman et al.). The ILAMB system integrates observational datasets spanning carbon, water, and energy cycles to assess model performance through quantitative metrics such as bias, root-mean-
square error, and temporal-spatial variability. The system is designed to be adaptable, accommodating various models or model versions while automating the benchmarking workflow. It is a model-agnostic framework constructed to intercompare and evaluate the results of any land or land-atmosphere model that produces the required output variables, making it scalable to accommodate a wide range of models (Collier et al., 2016, 2018b).The only requirement to include a model in the benchmarking is the format of the model output files, that must be in the Network Common Data Format (Unidata, 2025) and comply
with the Climate Forecast data conventions (Hassell et al., 2017). The automated benchmarking workflow generates a web document summarizing the statistical analysis[14]. ILAMB is designed to be used by all models and is employed by the GCP to compare model performances (Friedlingstein et al., 2023) but is also used by model groups directly. The LPJ-EOSIM team, for instance, runs ILAMB regularly for production-level runs.

### 3.4  Sensitivity analyses, MIPs, and hypothesis testing

For sake of completeness, we want to mention three more critical aspects related to model validation here. First, sensitivity analyses are useful to highlight various sources of uncertainty (Kleijnen, 2005; Saltelli et al., 2007). Second, model intercomparison projects allow us to understand the range of uncertainty across models, indicating where models diverge the most, thus showing where model communities can learn from one another and what processes require most attention (e.g., Frieler et al., 2024; Lawrence et al., 2016; Kou-Giesbrecht et al., 2023). Third, hypothesis testing allows modelers to pinpoint issues,
performance, and uncertainties of singular processes (e.g., Walker et al., 2017, 2021). A deeper investigation of these three topics will be helpful for modeling communities, but is out of scope for this paper.

### 4  Documentation - making the model understandable for users and developers

Documentation is essential for models, serving multiple purposes. It provides users with an overview when working with a model for the first time, guides developers in implementing new features, and describes the scientific processes behind the
model. Additionally, researchers outside the development team may seek detailed insights after reading a paper or wish to

---

[14]https://www.ilamb.org/results.html



run the model themselves. Various tools support effective documentation in the LSM community, from in-code comments and wikis to tutorials and even video guides.

## 4.1 Developer documentation

Proper documentation helps developers understand and maintain the code. It helps newcomers understand the model's archi-
tecture and functionality, reducing the initial learning curve, while ensuring consistency and quality of the evolving code. As described in Sect. 3.1.1, the first means of documentation must be the code itself, with proper naming of variables, functions, and classes, and modularization into understandable units. After this "self-documenting code", comments are useful for explaining complex logic, but their primary value lies in clarifying *why* something is done, or why it is done in a specific way. Additionally, comments should be used to provide scientific sources for methods or values. However, comments should be
a last resort after good naming and modularization prove insufficient clarifying steps on their own. More details on creating self-describing and testable code are in Gregor (2024b, a).

Even with clear code and comments, complex models can be difficult to grasp in their entirety. Therefore, tools that automatically generate documentation from the code itself and specially-formatted comments are crucial, creating HTML or PDF files that describe code units and their interconnections.

## 250 4.2 Scientific documentation

While developer documentation clarifies code functionality, broader documentation on model behaviors, assumptions, and caveats is essential for interpreting results. While ideally this information can be found in developer-focused documentation, this is often inaccessible or insufficient to non-developers or not extensive enough, so additional scientific documentation is necessary.

Publishing papers is one way to explain a model or new feature. Journals like *Geoscientific Model Development* and *Journal of Advances in Modeling of Earth Systems* are crucial because they provide venues for technical modeling papers, ensuring peer review and credit for development work. These journals allow more in-depth descriptions of model advancements than is normally acceptable in model application papers, allowing for improved model understanding and reproducibility. This is crucial since new features may take months to develop before they are applied in research.

However, publications alone are not enough. Open-access publishing may be infeasible for some teams which limits accessibility. Also, scattered literature can make information hard to find. Centralized, accessible documentation is ideal, for instance in the form of web pages. Web-based documentation, from simple GitHub wikis to dedicated websites, offers a user-friendly solution. It is searchable, linkable, and supports rich content like code snippets, equations, figures, and videos.

## 4.3 Usage documentation

While developer and scientific documentation clarify a model's technical and conceptual aspects, they often lack practical guidance for users. Most people interact with models as users, not developers, making clear user documentation essential. Key





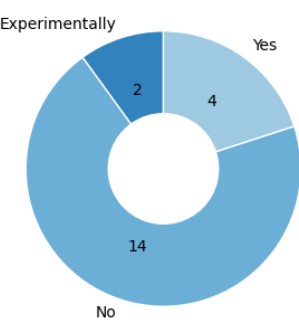

**Figure 3.** Usage of automated documentation tools is not yet heavily adopted in the LSM community. Four models use it actively while two are experimenting with it.

topics to include are installation, input data preparation, configuration, running simulations, and processing outputs. The better a model's usage is documented, the more people will use (and cite) it, and the lower the support burden as people can answer their own questions.

A well-structured README file is a crucial key entry point, providing an overview of the model, installation steps, usage instructions, and links to more detailed documentation. This is especially important for scientific workflows, outlining steps like pre-processing, model execution, and output analysis. Wikis on version control platforms (Sect. 5.2) offer more structured documentation, but for extensive content, a richer format is preferable. As with scientific documentation, webpages are often the best solution for user guides, as discussed in the next section.

## 4.4   Documentation tools

Automated documentation tools helps align the various documentation types. Six of the 20 GCP models actively use or explore such tools, suggesting potential for wider adoption (Fig. 3). Many programming languages already offer comment-based tools like PyDoc to generate HTML or PDF documentation from in-code comments. Furthermore, tools like DocTest (Foundation, 2024) combine documentation with testing. Here, *docstrings* describe expected outputs and serve as unit tests, failing if results
differ. While not all languages have this feature built-in, similar functionality can be achieved through external tools (e.g., in R: Hugh-Jones (2024)). Well-structured functions with clear purposes are easily tested and documented using these methods (see snippet 2).

   Doxygen[15] is a widely used tool for automatically generating documentation from source code, markdown, and image files. It transforms these into user-friendly formats like HTML and LaTeX, including descriptions of files, classes, functions, and
dependency diagrams. For scientific models, Doxygen offers valuable features such as equation rendering, figures, tables,

---

[15]https://www.doxygen.nl/



---

**Code snippet 2** An example of `doctest`, providing testing and documentation in one place. The comments explain what the expected outputs of the functions are and the doctest framework asserts that these are correctly computed by the function.

```python
def outgoing_longwave_radiation(temperature, epsilon):
    """
    This function calculates the longwave radiation that is emitted from a blackbody,
    depending on its emissivity epsilon and its temperature, based on Boltzmann (1884)
    >>> outgoing_longwave_radiation(0, 0.5) # no radiation at 0K
    0.0

    >>> radiation = outgoing_longwave_radiation(300, 0.5)
    >>> expected_rad = 229.635
    >>> math.isclose(radiation, expected_rad, rel_tol=1e-9)
    True
    """
    return epsilon * SIGMA * pow(temperature, 4)
```

---

citations, and modular organization for large projects. It also enhances traceability by extracting version control metadata like code revisions into the documentation. By integrating detailed explanations, code access, and visualizations, Doxygen improves collaboration, supports reproducibility, and promotes best practices in scientific programming. It supports multiple languages, using structured comments to include metadata like authorship and file details.

For Fortran code, FORD[16] offers similar features to Doxygen and is used by some of the models for code, scientific, and user documentation.

While automated documentation tools are highly valuable for technical documentation, other, semi-automated methods are more suitable for scientific and usage documentation, allowing simple editing and conversion to actual webpages. Sphinx (Sphinx, 2024) is another tool that can created complete, easily editable websites that can be published online, based on

ReStructuredText files (similar to Markdown, see also Wiggins et al. (2023)).

Keeping the documentation source files with the code makes it easier to update them alongside development, helping ensure that documentation remains up-to-date (see also Sect. 5.3). It also helps users find the documentation specific to the version of a model they are using. For example, the CTSM documentation[17] includes a drop-down menu for switching between documentation for different code versions.

Usability documentation can also be improved with published code notebooks, which combine formatted text (e.g., Markdown) with executable code and outputs. Available for languages like Python, R, and MATLAB, they can illustrate workflows such as plotting model outputs. Notebooks can also be converted into websites for public access (see Sect. 7).

---

[16]https://github.com/Fortran-FOSS-Programmers/ford

[17]https://escomp.github.io/ctsm-docs




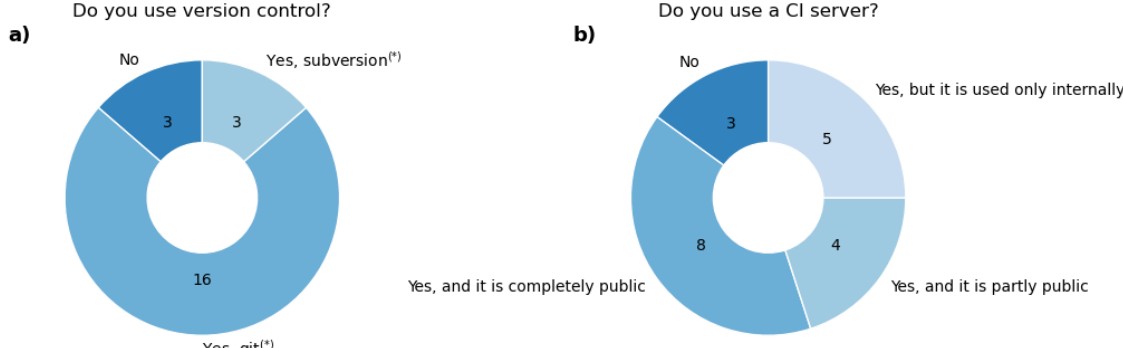

**Figure 4.** Survey results regarding version control and continuous integration (CI) within the 20 LSMs employed by the GCP. *) one model is currently using both git and subversion simultaneously and one model is in the process of moving from subversion to git.

## 5 Code and data management, version control, and continuous integration

### 5.1 Version control

Version control is the concept to manage and collaboratively work on code, allowing users to revert to previous versions of the code easily and to merge independent code developments together. There are multiple tools for this purpose such as Subversion or Git (e.g., Zolkifli et al. (2018)). Most (17) GCP models have a version control system in place, usually Git (Fig. 4). One model uses Subversion, while another model is using both Git and Subversion simultaneously, and one model is moving from Subversion to Git. Here, some key points of effective usage of version control are re-iterated.

Creating small commits with informative messages is the backbone of useful version control. This enables swift detection of bugs, by finding the first commit that breaks a test, e.g. with the `git bisect` tool. It also enables a better understanding of why each change was implemented, which is useful for code review and future development.

    New features should be developed on their own branches and re-integrated once development (including testing) is done. In that regard, it is advisable to divide these new features into small parts, allowing timely re-integration into the main codebase.

This prevents branches from lingering around for a long time, eventually making the re-integration more complex because the main codebase will likely have changed by then. This also helps code review (see Sect. 5.3) since extensive changes are harder to fully understand for parties not involved in their development, leading to poorer review outcomes. Since in science a lot of exploratory work is necessary and it may take years for a new feature to be developed in a scientific model, this fast re-integration might not be possible. One strategy to deal with this could be to include feature flags, allowing the inclusion

of new model code, but preventing its usage. A best practice is to continuously reintegrate new developments of the main code into the feature branch to keep development aligned, for instance by merging the main branch into the feature branch, or rebasing the feature branch.



Preprocessing workflows and statistical analyses of model outputs can become large and complex collections of code as well. These should also be put under version control, adhering to the same principles. A guideline on how to use Git in scientific workflows is given by Bryan (2018).

### 5.1.1 Traceability: versioning code and keeping track of experiments

Version control is the means of tracking code. But we also need to track entire experiments, to compare model or analysis code versions during development. Small commits and branching already help manage different code versions. Additionally, identifying the exact code version used in an experiment is crucial for both organization and reproducibility (Easterbrook and Johns, 2009). Git and Subversion `tags` serve this purpose by marking specific commits with a unique identifier.

For models still using Subversion, a common practice is to mention a revision number (e.g. `r1234`) when referring to the code used for a paper. However, the revision number does not always give enough detail about the code used for the paper, because it does not provide information on which branch was used. Therefore, a revision number needs to be combined with the name of the branch or tag.

Ideally, versioning employs the *semantic versioning concept* (Preston-Werner, 2013) and is integrated with the CI workflows, at least for new releases of a model, and to pinpoint the version of a model or workflow that was used for a paper. In any case, it is the responsibility of a scientist to adhere to versioning best practices, making sure that all code and data is committed and versioned for a model run, and that there are no local changes that are not accounted for.

### 5.1.2 Data version control

Apart from regular software, scientific workflows produce large amounts of data that need to be kept organized as well. Common tools to organize data alongside code are, for instance, Data Version Control[18] and the git large file system, git-lfs[19] which offers the possibility to store large files connected with git repositories. Some models use versioned S3 buckets, and Zenodo for larger one-time data dumps (especially after publication), while some have Subversion servers. Another solution within the community is using netCDF files with version information in metadata. However, model representatives agree that data version control is a core issue, yet there are no clear best practices from the community that others could adopt as of now.

### 5.2 Continuous integration

Continuous integration (CI) is the practice of integrating new code frequently into the code base whilst verifying it through automated tests (Meyer, 2014; Fowler and Foemmel, 2006). CI is usually done on a platform like GitHub, Gitlab, or others. These offer easily accessible interfaces to monitor code status, track ongoing developments, and facilitate reintegration into the main codebase. This approach keeps developers aligned, visualizes progress, and ensures code correctness. It also allows for additional features such as hosting documentation, archiving code, and managing a project. Some ideas and tips how to use CI in science are given by Braga et al. (2023).

---

[18]https://dvc.org/
[19]https://git-lfs.com



Notably, CI offers *pipelines* to pass code through various stages, for instance, compilation, formatting, automated testing, and documentation generation (see Fig. C1). CI relies on version control systems like Git, where new features, bug fixes, or
improvements are developed on separate branches. Once the developer is satisfied with their work, they initiate a *merge request* (also called *pull request*) to integrate the changes. This process fosters collaboration through a web interface, where developers can review, comment, and refine the code before merging it into the main branch.

Of the GCP models, we found 12 to have some automated code pipelines, mostly including code formatting, compilation, and simple tests. Three models have more advanced CI setups, including larger test suites, linting (see next section), automated
bitwise comparisons, and updates of automated documentation. LPX-Bern and CLASSIC, for instance, also have container-based pipeline steps for plot comparisons. Differences of key variables between new output and a reference version are created, enabling a quick comparison of results. We therefore suggest that there are already some notable examples, but in general there is some room for modeling communities to improve their CI processes. One notable example from the community is the usage of buildbot within the ICON-Land model.

### 5.2.1 Buildbot for ICON-Land/JSBACH

Buildbot[20] is an open-source continuous integration (CI) framework for distributed, parallel job execution across multiple platforms. It is used in the development of the ICON weather and climate model (ICON partnership (MPI-M; DWD; DKRZ; KIT; C2SM) (2024), Hohenegger et al. (2023)), including its LSM ICON-Land/JSBACH, to ensure code changes do not break the model before being merged. Tests are triggered manually via GitLab merge request comments, initiating automated
pipelines on Buildbot servers hosted at DKRZ (German Climate Computing Centre).

Tests run in parallel across various systems, from Linux workstations and Mac computers to high-performance computing (HPC) environments using CPU, vector, and GPU architectures. Multiple compilers and build options are tested, ensuring over 50 different builds for model portability. Notably, this requires having such architectures available and connected to buildbot.

Each full test suite runs hundreds of model experiments. Since this comes with a non-negligible cost factor, the test simula-
tions need to be relatively short but still cover large and diverse parts of the code, including standalone and coupled experiments for atmosphere, land, and ocean components. Bit-identity of results is tested with respect to reference data from the previous model code as well as different runtime configurations such as restarts and varying process/thread counts. GPU experiments ensure results remain within acceptable tolerances compared to CPU runs.

Note that similar setups can be configured with other CI-frameworks like Gitlab or Github Actions as well.

### 5.2.2 Code formatting and static code analysis

Programmers often debate whether to use tabs or spaces, but what matters is consistency. Modeling communities should adopt standardized code style guidelines, following language conventions (e.g., `snake_case` in Python, `camelCase` in Java). These should be enforced via code formatters integrated into CI pipelines and local development, such as pre-commit hooks,

---

[20]https://buildbot.net/



which automatically correct formatting before commits. Usually, the piece of software that checks the format can also apply
the correct format automatically.

Consistent code style is more than aesthetics because it prevents minor differences (e.g., whitespace changes) from cluttering
version history and obscuring meaningful edits which impair code review. It also improves readability by making variable roles
(e.g., objects, functions, constants) immediately clear (see also Gregor (2024a)).

Modern programming languages often provide standard formats and formatters, such as `rustfmt` for Rust, while others
define style guides like PEP8 for Python. Instead of debating an ideal style (an inherently subjective and time-consuming task),
communities should adopt established standards, ensuring familiarity for both internal and external developers. Switching
to a consistent style in an existing codebase can be challenging. Reformatting all files at once risks conflicts when multiple
developers work on different branches. A gradual approach—enforcing style only on modified files—helps integrate formatting
without disrupting active development.

Static code analyzers format code ("linting") and use static analysis to detect issues like unused imports, missing error
handling, or potential bugs like type violations or out-of-bounds access. Their effectiveness varies by programming language,
with statically typed languages benefiting from more thorough analysis. Some compilers also include static code analysis,
contributing to improved code quality and maintainability. The most commonly used tools for the LSM-languages are *clang-
tidy*[21] for C++, and *F-Lint*[22] for Fortran. The latter, however, is not freely available. Free alternatives for static Fortran code
analysis are, e.g., *fortran-src* (Contrastin et al., 2025) and *FortranAnalyser* (García-Rodríguez et al., 2024). Notably, some
languages that are not statically typed per se (meaning, the concrete type of the variable does not need to be specified in
advance, like in Python) offer tools like type hints to indicate the type of a variable (see snippet 1).

## 5.3  Code review and following best practices

A version control platform can ensure that a new feature is ready for integration by handling tasks like compiling, linting,
and testing. However, aspects like logical correctness, guideline adherence, and documentation updates often require manual
review. In professional software development, code reviews are an absolute standard, typically conducted within the version
control platform, which provides an interactive, transparent discussion space. No code will be integrated into the codebase
without a review. The process is often guided by a checklist defining feature completeness, a practice that could also benefit
model communities (see Sect. C1).

Code review practices within the LSM community vary widely, from voluntary to mandatory reviews with differing levels
of rigor. Some models follow well-defined guidelines, protocols, and checklists before reintegration. Some use issue[23] and pull
request templates[24] to structure the review process. Several models explicitly require feature flags to ensure new additions do
not alter existing results. A few, like ELM, mandate specific testing and documentation, such as adding at least one integration
test to nightly test suites and standalone documentation for each new feature.

---

[21]https://clang.llvm.org/extra/clang-tidy
[22]https://codework.com/solutions/developer-tools/f-lint/
[23]https://github.com/ESCOMP/CTSM/tree/master/.github/ISSUE_TEMPLATE
[24]https://github.com/ESCOMP/CTSM/blob/master/.github/PULL_REQUEST_TEMPLATE.md





Thorough code reviews increase development time upfront but prevent costly issues later. Group leaders in model communities should foster a culture that values technical quality and recognize review contributions, such as in model release notes. Reviews can also help junior members familiarize themselves with the code. Additionally, pair programming, where two developers collaborate at one workstation (Williams, 2010), can improve code quality and knowledge sharing, though it may extend development time (Hannay et al., 2009).

A critical issue here is once again the limited number of people available to review, their limited time, and the lack of credit (see Merow et al. (2023)). In small model communities, finding reviewers is particularly challenging. Only few models have dedicated scientific programmers, yet these roles are crucial for maintaining technical standards and allowing scientists to focus on the science. This shortage is partly due to funding limitations, especially for smaller models or those not tied to climate models, which cannot afford to hire scientific programmers.

Notably, large language models (LLMs) enable automated code reviews. Tools like Coderabbit[25] combine traditional linters with LLMs to suggest code improvements, from individual statements to large refactorings. Within a merge request, developers can refine these suggestions, allowing the system to learn project-specific preferences. This can accelerate reviews by catching obvious issues and providing targeted recommendations. However, reviewers must carefully validate LLM-generated suggestions to avoid incorporating suboptimal or incorrect solutions. Notably, the LLM does not execute the code and it can happen that it hallucinates bugs which may require the developer to be quite technically versed to understand they are wrong. Since model developers often do not have a solid background in software engineering, we urge to use great care with such tools.

## 5.4 Continuous Deployment

Finally, in software engineering, CI is often paired with CD (continuous deployment/delivery), meaning that new features are automatically rolled out to users soon after development once they have passed all required pipeline stages such as review and testing. This aspect is likely not relevant for modeling communities, as new model versions are not shipped to users. Rather, a slow release process, possibly including a model development publication, will happen. However, CD could become important for visualization tools built on top of models that aim at making science publicly available. We will touch upon this in Sect. 7.

## 6 Reproducibility, usability, and portability of scientific workflows and results

In recent years, science has been described as facing a reproducibility crisis affecting nearly all scientific disciplines (Baker, 2016). While the gravity of the problem, or at least the terminology, is debatable (Fanelli, 2018), scientific experiments are often becoming increasingly hard to reproduce. Being pieces of software, model runs and their analyses should be easily reproducible, but this is not always the case.

In a first attempt to tackle this issue, efforts have been made to make scientific results more openly available, for instance through the FAIR data principles (Wilkinson et al., 2016). Also models are increasingly becoming FLOSS projects. However, most scientific models do not adhere to this idea yet (Barton et al., 2022), and accessibility remains a major challenge in Earth

---

[25]https://www.coderabbit.ai





system science (e.g., Añel et al., 2021). As elaborated above, most LSMs of the GCP are openly available through GitHub or a similar entity while others have some model versions open to the public through repositories like Zenodo or make them available upon request. But FAIR principles are not enough. Even if everything were openly available, it would still be very hard to rerun a simulation, let alone in combination with data preparation and post-processing.

Tremendous amounts of data, installation of software, the complexity of environment set-up, the combination of multiple tools, languages, scripts, processing steps and models, and computational or time constraints prevent other scientists from reproducing the results of others, and thereby building on existing work (Easterbrook, 2010). But they also make it harder to collaborate in the first place. One solution how this can be addressed are shared computing systems[26] or cloud-based services like AWS S3 in combination with ParallelCluster. These offer the possibility to easily copy entire setups between users or

provide users access to set-up systems without the overhead of installing libraries themselves However, this does not address the problem when people outside of the same institution or project want to re-run an analysis or build their work on it.

To address the larger issue of reproducibility and portability, Mölder et al. (2021) have defined the term *sustainable data analyses* for data analyses that are transparent (enabling assessment of the methodological validity), reproducible (ensuring computational validity), and adaptable (ensuring reusability). Beside the validation aspects, sustainable data analyses offer an

additional benefit for the scientific community and the original authors to build upon them or modify them for new research questions, in contrast to being only usable for a single publication. Six aspects are relevant in that regard: automation, scalability, portability, readability, traceability, and documentation. Readability and documentation and in parts traceability were already addressed above, which not only applies to the model but also any processing or data analysis parts. In the following, we will address how to achieve traceability, portability, scalability, and automation.

## 465 6.1 Automation, traceability, and reusability of scientific workflows

While geoscientific models are complex themselves, a tremendous amount of complexity also comes from the entire workflows around it, from pre-processing of data over model runs to post-processing and data analysis. All previously mentioned paradigms also apply to such entire workflows. For the automation, traceability and reproducibility of such a workflow, there are workflow management systems (or "pipeline tools") like `targets` (Landau, 2021a) or `Snakemake` (Mölder et al., 2021).

These allow combining multiple heterogeneous steps such as pre-processing scripts, model runs, post-processing, and visualization into combined, reproducible workflows that are written in code. They can then execute these workflows, thereby checking whether code or inputs have changed and only running those parts that are affected by the changes. Such an automated setup not only helps streamline the process but is also critical in avoiding non-systematic errors from manual steps, while allowing every step to be rerun, tracking all intermediate outputs and keeping everything aligned. Furthermore, the tool

PEcAn embraces the same principles, but in a multi-model context Fer et al. (2021).

---

[26]e.g., https://jasmin.ac.uk



### 6.1.1 Snakemake

Snakemake is one of the most widely used workflow management systems in sciences, but it is not yet common in earth or environmental sciences (Mölder et al., 2021). Snakemake workflows are defined by so-called rules defining individual steps in terms of their input, output, parameters, and how the output shall be obtained from the input (e.g., by running a command line application, a script or a notebook). The rule definition happens in a domain-specific language, enabling rules to be enriched by arbitrary Python logic (e.g. for complex aggregations, parameter retrieval, and configuration). Various languages are supported for individual workflow steps.

Via so-called wildcards Snakemake allows the easy processing of entire parameter spaces—for instance for different regions, resolutions, or input scenarios. In combination with Git branches and data version control, one can also handle runs of various versions of the same model, which is a vital aspect in the process of geoscientific model development and geoscientific studies (Easterbrook and Johns, 2009).

From the given rules, Snakemake infers a dependency graph. Independent parts of the graph can be processed in parallel, allowing fine-grained control over resoureces like CPUs or memory. Through plugins[27], Snakemake workflows can be executed on local machines, compute servers, clusters (Slurm, LSF, etc.), and cloud middleware, while utilizing various kinds of storage (local, NFS, S3, etc.). Importantly, a Snakemake workflow can be scaled to any of these infrastructures without modifying the workflow definition itself, thereby avoiding a lock-in effect to the setup of the original authors.

To ensure reproducibility, it is important to make sure that every workflow step is computed on exactly the intended software stack (i.e., required tools, libraries, and versions thereof). Snakemake integrates with the programming language–agnostic package manager Conda, allowing isolated software environments to be defined per analysis step. Alternatively, analysis steps can be executed within containers (see Sect. 6.2.2).

After processing a Snakemake workflow, it can automatically generate interactive HTML reports that link results with the parameters, code, and software used. These reports provide comprehensive and traceable supplementary materials for published manuscripts.

Mölder et al. (2021) provide an extensive explanation and examples of how to use Snakemake in scientific workflows, while we provide an example around the model LPJ-GUESS (see Sect. 7 and Figure 5).

### 6.1.2 targets

Like Snakemake, the targets R package is a workflow management system. It brands itself specifically as a pipeline toolkit intended to facilitate reproducible research with computationally intensive data analysis (Landau, 2021b). As such it can be a useful tool for pre-processing data and data analysis in the context of geoscientific modeling studies.

The core feature of targets is its tracking of individual workflow steps, the eponymous targets, using a directed acyclic graph to detect dependencies. At runtime, this allows the pipeline to determine which targets are out-of-date and must be run and which are up-to-date and can be skipped. This not only reduces costly, redundant computations but also shifts the overhead

---

[27]https://snakemake.github.io/snakemake-plugin-catalog





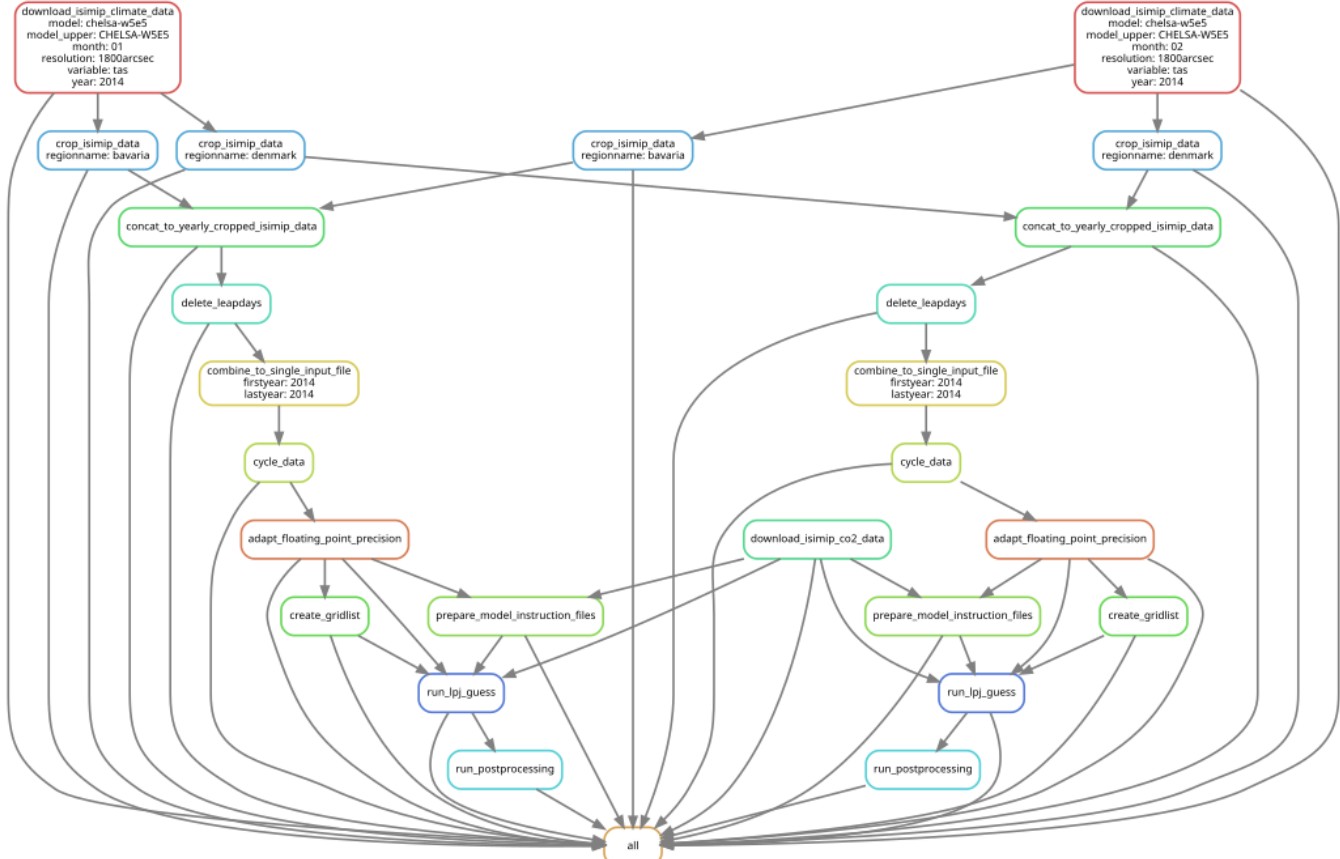

**Figure 5.** Illustration of the Snakemake workflow from our showcase. The pipeline consists of downloading, cropping, and mapping data to enable a model run with the LPJ-GUESS model. Different regions can easily be defined, resulting in one model run per region, in this case for two regions. Snakemake takes care that when an intermediate result changes, all steps depending on this intermediate result will be re-executed.

from programmer to program. Each time the pipeline is successfully run the programmer can be virtually certain that the desired output data or analysis results match the underlying code.

Crucially, targets is designed to enable scalable workflows and (relatively) seamlessly integrates with HPC. Based on the dependency graph, targets is able to identify individual steps which can run in parallel allowing for efficient use of HPC. This dependency graph can also visualize, for instance, resource consumption of each step. Because targets is agnostic in regards to 'where' the individual processes are executed, the pipeline itself need not be altered to be scaled, for example, from a local machine to an HPC platform. Rather, the user must only specify the appropriate backend that the workers should be executed

on for a given use case.



Targets is explicitly designed for the R programming language and intended for use with a function-oriented programming style. Rather than individual targets executing entire scripts, they execute a single function with, ideally, a clearly defined expected output (e.g. a single dataset, no hidden side-effects).

### 6.1.3 PEcAn

The Predictive Ecosystem Analyzer (PEcAn) is a model-data informatics system designed to make model usage more accessible, transparent, and repeatable (Fer et al., 2021; Dietze et al., 2013). PEcAn supports >20 land models, ranging from simple single site models, to global vegetation models. PEcAn can call any model through a common interface and uses a common output standard to make model analyses scalable. PEcAn workflows can be triggered through a web-based graphical interface, an API, or directly within R code, and executed locally or remotely, through direct execution, a HPC queue, or in the cloud via a Kubernetes stack of Docker containers. PEcAn also comes with a diversity of automated, scalable pipelines for transforming various model inputs (model parameters, meteorology, soils, vegetation, phenology, etc.) into PEcAn standard, performing gap-filling and downscaling as needed, and then converting these inputs into model-specific formats.

PEcAn extends versioning to the larger analysis workflow, including the option to use a provenance-tracking database that assigns each modeling workflow, and each run within a workflow, a unique traceable ID to keep track of model inputs, analysis outputs, and other settings, which can be helpful

PEcAn places a particular emphasis on model-data integration and uncertainty quantification. As such PEcAn is designed to run model ensembles by default, with single model runs being an ensemble of size n=1, and provides tools for the propagation and partitioning of model uncertainties (parameters, drivers, initial conditions, boundary conditions) (LeBauer et al., 2013). To reduce uncertainties PEcAn also provides model parameter calibration tools, including an emulator-based multi-site hierarchical Bayesian calibration (Fer et al., 2018), and tools for ensemble-based iterative data assimilation, at both the site and continental scales, for the purposes of state-variable estimation (a.k.a. reanalysis) and automated near-term forecasting (Dokoohaki et al., 2022). The latter includes additional automated input pipelines for ingesting the various bottom-up and remotely-sensed data constraints used in the data assimilation system. Current PEcAn development is focused on the integration of process/ML hybrid approaches, including downscaling, bias correction, and modeling spatial parameter heterogeneity, as well as applications of PEcAn to GHG monitoring, reporting, and verification (e.g., both Finland and California employ PEcAn as part of their carbon accounting systems).

## 6.2 Portability

Ensuring traceability does not guarantee that code can run on different platforms, but portability is crucial for building on existing work. Nonetheless, there are multiple options to achieve portability as outlined below. Notably, bit-by-bit equality might not be achievable across environments ("the default assumption should be that [Earth system models] are not replicable under changes in the HPC environment", Massonnet et al. (2020)). Portability issues can arise from various sources, including coding flaws (for instance, not properly initializing variables), compiler optimizations (e.g., fast-math), different compilers, or non-deterministic/numerically unstable libraries. While not all discrepancies can be eliminated, model developers should strive





---

**Code snippet 3** Excerpt of the `environments.yml` exported from the `conda` environment of our showcase, containing information about the used libraries, their versions, and build identifiers.

---

```yaml
name: model-coding-paper
channels:
  - conda-forge
  - bioconda
  - nodefaults
dependencies:
  - cartopy=0.24.0=py312hf9745cd_0
  - cdo=2.4.4=h1f03bf2_1
  - geopandas=1.0.1=pyhd8ed1ab_1
  - matplotlib=3.9.2=py312h7900ff3_2
  - netcdf4=1.7.2=nompi_py312ha728dd9_100
  - numpy=2.1.3=py312h58c1407_0
  - pandas=2.2.3=py312hf9745cd_1
```

---

for portability across systems, ensuring statistically consistent results between systems to maintain scientific reproducibility.
A first critical step is running identical simulations on multiple environments and analyzing result variations to establish a baseline for assessing portability.

### 6.2.1 Package versions

To enable the running of entire workflows on various systems, tools like conda, capsule, or Renv, come in handy. These allow definitions of the exact versions of libraries that should be used. On a new machine, the same environment can then be created
from this definition, enabling a re-run of the workflow using the same library versions, thus assuring, in the ideal case, identical results (see snippet 3).

### 6.2.2 Making models portable with containers

Container images are standardized software units that bundle code with its dependencies, ensuring that an application runs efficiently and consistently across different environments. Container images run in containers, which are an isolation mechanism that is similar to, but more light-weight than virtual machines. For models, a container image could contain the necessary inputs, model code, libraries, and post-processing scripts. A user can then run the model and all post-processing and visualization without having to install anything on their computer (apart from the container software itself).

Such portable solution are invaluable for repeating analyses (both to test reproducibility and to repeating analyses after fixing bugs), but also for extending them, for instance for additional sites or regions, or for updating analyses over time (which is a
key aspect of the GCP), or for example that can be built upon by other groups. Some models such as CLASSIC and CTSM





already offer this functionality, while multiple models are working on it. Reviewers might also be interested in re-running a case study to get a better idea of the work under review, for instance by also checking outputs that did not make it into the paper.

Already a decade ago, Docker was suggested by Boettiger (2015) as helpful tool in making science reproducible. Its adoption
in the LSM community remains limited but some LSMs have been "containerized" with Docker (e.g., Keetz et al., 2023; Lombardozzi et al., 2023). Besides facilitating portability and installation, a major advantage of such container solutions is that they can include additional code and software dependencies necessary to reproduce the broader modeling pipeline. For example, the Docker-based Land Sites Platform (LSP; Keetz et al., 2023) enables single-site simulations with the demographic vegetation model Functionally Assembled Terrestrial Ecosystem Simulator (FATES) embedded in CTSM as the host land model. Because
CTSM-FATES single-site simulations have relatively modest hardware requirements, the LSP can run experiments without the need for HPC infrastructure in its standard configuration. The container also provides the required input data (land surface information and climate forcing) for a set of example sites. Accordingly, users do not need to download large datasets and experiments can easily be reproduced on common computers such as personal laptops. Importantly, the LSP containers also include the software dependencies and educational example code to create new input data (but this requires access to global
datasets) and analyze the model outputs. While primarily designed as an educational tool to foster interdisciplinary collaboration, the LSP therefore also enhances LSM reproducibility and accessibility. Similarly, the PEcAn project maintains Docker images for many of the individual LSMs. This enables PEcAn to launch multiple model container instances (e.g., in the cloud) and provide inputs and configurations for individual runs (e.g., within a model ensemble or for multiple sites), and centrally collect the model outputs for analysis.

Apptainer is a tool that offers similar functionality as Docker and is also used within the LSM community. CLASSIC, for instance, provides a site-level benchmarking suite using Apptainer (Melton et al., 2019a, b) which included the model dependencies needed to run CLASSIC at site scale. A benchmarking suite allows users to run 31 FLUXNET2015 sites and reproduce the figures of the paper describing CLASSIC v. 1.0 (Melton et al., 2020). This presents a reproducible, portable and reasonably low-barrier method to evaluate the model[28].


## 6.3 Visualization and usability of scientific methods and results

Since Earth system modeling and related fields are dealing with issues that are highly relevant for the public, it should be a key aspect to make these scientific results available to the public, and not only in a scientific paper. Policy advisors and journalists
will be interested in exploring model results. Therefore, tools to easily view model outputs are helpful. These include, for instance, shiny-apps (R), Jupyter notebooks (Python), and Snakemake reports. A good compromise between usability and effort is critical. Ideally, the tool should also be helpful to the scientists to explore their results. As discussed in Sect. 3.3, part of the validation process is investigating plots, for instance of global maps of a variable. So, efforts to facilitate such

---

[28]https://cccma.gitlab.io/classic_pages/info/get_started/





investigation can be helpful both for the modeling community and the broader audience at the same time. In our showcase we
show a simple example of achieving this and a notable example from the LSM-community is the ORCHIDEE visualization
system "MAPPER"[29], see Sect. A1.

## 7   A showcase

To exemplify the discussed aspects, we provide a full processing workflow[30] that can help as a starting point for other modeling
projects. This example can be used by modeling communities as an inspiration for their own workflows. It contains a simple,
but typical pipeline of an LSM (in this case, LPJ-GUESS), including data preprocessing, post-processing of model outputs,
and publicly hosted interactive plots with minimal effort. The repository contains:

- An `environment.yml` to allow to install all packages in the correct versions to re-run pre- and post processing

- A `Snakemake` pipeline to re-run the entire workflow on a different machine. This includes data pre-processing, a full
  model run of LPJ-GUESS as a Docker container, and simple post-processing and data analysis with Jupyter notebooks.
This workflow is executed twice, for two different regions (Bavaria in Germany, and Panama), and can be easily con-
  figured to be re-run for an arbitrary world region, at a different resolution, for a different time period, highlighting the
  adaptability of such a workflow

- A `Dockerfile`, showing how an LSM can be published as a docker image, to facilitate running it on any computer
  without any required installation

- Some `Unit tests` of the data processing and analysis code

- A `Github Actions` pipeline running automated code cleanup (linting), unit tests, compilation checks, as well as
  the entire `Snakemake` pipeline (including running LPJ-GUESS in a Docker container on GitHub server and pre- and
  post-processing).

- Continuous deployment for interactive plots: Using Jupyter, plotly, and GitHub pages, interactive plots of the model
outputs, including geographical maps and animations, are made publicly available[31]

The full run within the GitHub Actions pipeline already showcases that the pipeline is completely portable, and reproducible.
Only Python and Docker need to be installed on the machine. GitHub Pages, Plotly, and Jupyter make it easy to host interactive
plots online for anyone to explore. While other tools could accomplish the same goal, we chose these for their simplicity and
minimal setup effort.

---

[29]https://orchidas.lsce.ipsl.fr/mapper/

[30]https://github.com/k-gregor/modeling-software-tools

[31]https://k-gregor.github.io/modeling-software-tools/



# 8   Conclusions

In this paper, insights from professional software engineers and the modeling community were introduced to derive some guidelines on how modeling communities can improve their workflows, make their code less error-prone, more understandable, and more reproducible. Scientific modeling is software development, but with very particular constraints. Therefore, not all software engineering principles will be applicable. Nonetheless, for scientists it will be helpful to know about these tools, frameworks, concepts, and principles, to improve their everyday work and their models, to understand where model development and workflows can be improved. The limited available time, training, and scientific credit for solid programming remains a point of concern in model development. We would therefore like to stress the importance of scientific programmers in terms of solid model development and thus solid science. Furthermore, we argue that workshops or trainings for programming in geoscientific modeling will be helpful. Our paper can hopefully serve geoscientists by providing ideas, discussions, and examples of good practices and can hopefully be used as a valuable resource for various modeling communities.

*Code availability.*   The code for the showcase is available at https://github.com/k-gregor/modeling-software-tools. The exact version of the showcase described in this paper is archived on Zenodo under DOI `10.5281/zenodo.15191115` (Gregor, 2025)





**Appendix A**

**A1    Examples from the LSM modeling community**

In the following, we list some notable examples of model testing, validation, and visualization from the community that may serve as inspiration for other modeling groups.

**A1.1    DGVMTools**

DGVMTools is an R package designed for processing and analyzing LSM output and associated data. Unlike some of the examples here, it is not a benchmarking suite and it is not focussed on a specific model. Rather, it is a library of functions
from which verification and validation scripts can be built. It implicitly supports the spatial and temporal dimensions found in LSM but does not require gridded data. It features NetCDF file format to enable compatiblity with many models and datasets, and includes bespoke functionality to read output from the LPJ-GUESS, aDGVM and aDGVM2 models. This flexibility, combined extensive nature of the R language, means that DGVMTools can be used for a wide range of tasks to support model development, ranging from benchmarking against site measurements to evaluation of model performance at global scale.

**A1.2    AMBER**

The Automated Model Benchmarking R Package (AMBER[32]; Seiler et al. (2022)) was developed to evaluate CLASSIC using a skill scoring system based on ILAMB, whereby scores are normalized between 0 and 1 for five metrics of model performance. AMBER allows a model to be evaluated against point-scale reference datasets in addition to gridded products. When multiple observation-based reference datasets are available for a single variable, AMBER scores each dataset relative to the
others leading to a 'benchmark' score. If a model has a score greater than the benchmark score, it is considered statistically indistinguishable from an observation-based reference dataset. This approach thus takes into account the uncertainty associated with observation-based datasets. AMBER can be triggered to run at the end of every CLASSIC simulation, ensuring a regular, routine evaluation against a suite of reference datasets and variables (58 reference datasets across 24 variables as of January 2025). Notably, AMBER is designed so that it can be used by all models.

**A1.3    lpjmlkit and lpjmlstats**

The *lpjmlkit* R package (Breier et al., 2025) provides functionality for streamlining the scripted set up of single or multiple sets of LPJmL simulations, allowing direct access to model runtime settings as well as model parameters and the definition of run dependencies (e.g. run A needs to finish before run B and C are started). The package also provides data classes and related functions to work with LPJmL input and outputs files, including automatic processing of extended meta data. The
*lpjmlstats* R package (Hötten et al., 2024) is built on top of *lpjmlkit* and provides statistical tools for LPJmL data analysis. Its main functionality is a benchmarking tool that allows visual comparison between multiple LPJmL runs in terms of tables, time

---

[32]https://cccma.gitlab.io/classic_pages/benchmarking/





series graphs and maps. By default, the benchmark creates a PDF report but also returns the data shown in the report to allow for custom visualisation or further analysis. The benchmarking is designed in a modular fashion, allowing for the definition of different benchmarking metrics that can then be applied to groups of outputs.

### A1.4 LPJ-GUESS Delta Reports

The delta report script of LPJ-GUESS generates a difference-report from benchmark output reports from LPJ-GUESS, allowing to visualize differences in output variables between model versions.

### A1.5 MAPPER (ORCHIDEE visualization system)

The MAPPER[33] is the software utility and web-site used to visualize ORCHIDEE simulation results. This tool is actively used by the ORCHIDEE community to track the model evolution, rapidly evaluate new developments or parameter optimization, and compare modeled variables versus observations. It plots global maps and time-series for pre-selected regions for key variables of several categories (Energy, Water, Carbon, etc.), creates difference plots between various simulations or versus data products and offers sophisticated visual diagnostics (e.g.: river discharge and river basin precipitation graphs, scatter plots between different model variables). Each major ORCHIDEE revision corresponding to a certain model development stage is first plotted and analyzed through the MAPPER.

### A1.6 ORCHIDAS (ORCHIDEE Data Assimilation System)

ORCHIDAS[34] is the data assimilation tool developed for optimization of the key ORCHIDEE parameters controlling the carbon, water and/or energy budget using various data sources (e.g. in situ measurements, satellite products, inventory data or expert knowledge on the physical ranges of variation of modeled variables). ORCHIDAS is used during development for validation of simulation results versus various observational products, but even more than that for the tuning of ORCHIDEE parameters in order to have a better fit between simulations and observations. Parameter tuning is currently implemented in ORCHIDAS using multiple approaches including gradient-based (L-BFGS-B algorithm, Byrd et al. 1995), Monte-Carlo random-based (Genetic algorithm, Goldberg 1989), 4D-ensemble variational (LaVEnDAR approach, Pinnington et al. 2020) and statistical methodology (History Matching, Williamson et al. 2013). Sensitivity analysis can also be run which is used to test the impact of a given parameter on a given output. Initially developed for the ORCHIDEE model, the ORCHIDAS tool has been also adapted for CTESSEL (land component of the ECMWF integrated forecasting system).

### A1.7 ORCHIDEE trusting

ORCHIDEE trusting[35] is a test suite performed for the ORCHIDEE model on a nightly basis. It is run using different compile options and run modes (coupled with the atmospheric general circulation model LMDZ or forced by atmospheric forcing files)

---

[33]https://orchidas.lsce.ipsl.fr/mapper/

[34]https://orchidas.lsce.ipsl.fr/

[35]https://webservices.ipsl.fr/trusting/





on multiple platforms. Simulation results (restart files, some diagnostics and certain outputs) are compared bit-by-bit with the previous version of ORCHIDEE. If a certain commit leads to the model dis-functioning or an involuntary change in the results, it is automatically revealed by the trusting tool and the model is reverted back to the previous revision. The trusting also keeps track of the computing time spent on each test, which can be very useful for analysis of the model evolution when a significant change in its productivity is observed.

- Eventuell könnte man die Reihenfolge der Beispiele noch mal ändern. Scheint jetzt in der Reihenfolge zu sein, in der die Leute es eingetragen haben. Vielleicht stattdessen alphabetisch? Oder vielleicht "DGVMTools" als letztes, da es im Gegensatz zu den anderen Beispielen nicht speziell für ein Modell ist? - Eventuell könnte man die Reihenfolge der Beispiele noch mal ändern. Scheint jetzt in der Reihenfolge zu sein, in der die Leute es eingetragen haben. Vielleicht stattdessen alphabetisch? Oder vielleicht "DGVMTools" als letztes, da es im Gegensatz zu den anderen Beispielen nicht speziell für ein Modell ist?

## Appendix B

### B1    Survey of coding practices within the LSM community

To understand the current state of software development in the LSM community, we sent out a survey to all 20 land surface models of the Global Carbon Project. We asked the following questions:

### B1.1    Survey questions

What is the name of your model?

What is your name and email?

What is the main programming language of the model?

Does your model use a version control system?

– No

– Yes, git

– Yes, subversion

– Yes, mercurial

– I don't know

Does your model code contain unit tests (automated tests that test certain parts of the code)?

– No

– Yes, there are a few unit tests

– Yes, the model has an extensive unit test suite



– I don't know

Does your model employ runtime assertions / checks (e.g. checking whether values are within an expected range, checking

whether mass balances are fulfilled)?

– No, there are no checks evaluated at runtime

– Yes, some critical checks are done

– Yes, runtime assertions are used throughout the model's code base

– I don't know

Does your model do any bitwise comparison tests when a new feature is added?

– Yes

– No

Do you use a continuous integration server, meaning, is the code hosted on a platform like GitHub, Gitlab, TravisCI, CircleCI,

Jenkins, etc.?

– Yes, and it is partly public

– Yes, and it is completely public

– Yes, but it is only used internally

– I don't know

If you do have a continuous integration server that is at least partly public, please provide a link here:

What actions are performed within the continuous integration pipelines, e.g., after a commit? Examples: compile checks, code

formatting, automated testing, automated documentation, ...

Do you use a tool for automated generation of documentation?

Do you have some sort of an automated benchmarking setup?

To your knowledge, has the model previously been provided with pipeline code to enable reproduction of scientific results

(e.g., using tools such as Snakemake or Docker)? Or has this been discussed in your community?

Can you provide examples of model intercomparison projects which your model has participated in?

Is there functionality to make the model usable by non-scientists, for instance, a web-based version, or an education version?

Do you have any final remarks you would like to share? Any tools that you use for your model that others should be aware of?

**Appendix C**





Keeping such a checklist and referring to it in the process of merging new code helps making sure that nothing is forgotten, for instance, not having looked whether any documentation needs to be changed.

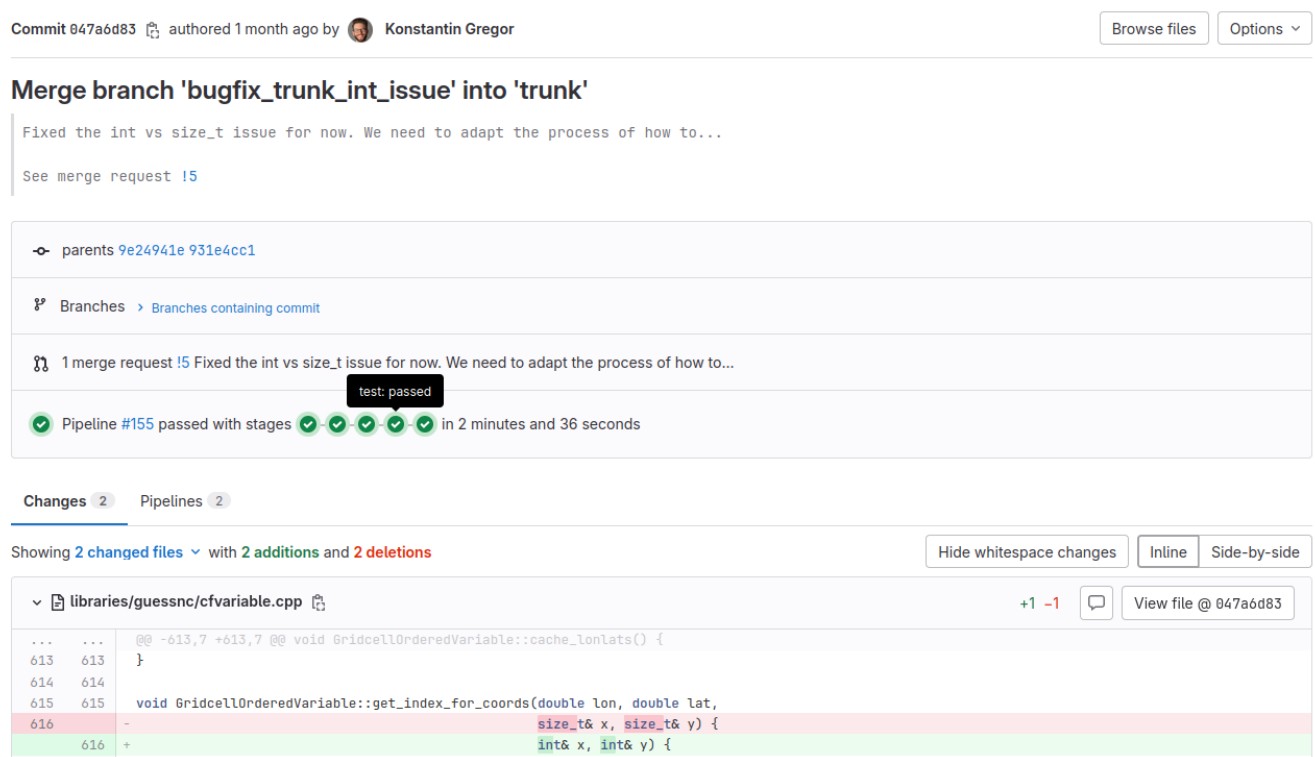

**Figure C1.** Example of a merge request on a continuous integration server, here on Gitlab for the LPJ-GUESS model: code changes are made visible to other developers with the possiblity to comment on them. Pipelines are run with multiple steps including compilation and testing.



*Author contributions.* KG conceived of the paper, conducted the survey, created the visualizations, and wrote the manuscript. BM, TG, VJV,
MF, JD, and AR helped in conceptualizing the paper. All authors contributed in writing and editing of the manuscript.

*Competing interests.* One of the authors, SR, is on the editorial board of GMD.

*Acknowledgements.* We would like to thank Eva Lieungh, Jinxin Liu, Anthony Walker, Will Wieder, Xu Yue, and Sönke Zaehle for filling
out the survey. We acknowledge the usage of chatGPT to aid with language editing. Specifically, chatGPT was asked to rewrite some original
paragraphs of the manuscript with clearer wording. KG acknowledges funding from the VELUX Stiftung (project 1897, www.veluxstiftung.
ch). JK thanks ACCESS-NRI for supporting code development and infrastructure management related to the CABLE-POP model. This
manuscript has authors at Lawrence Berkeley National Laboratory under Contract No. DE-AC02-05CH11231 with the U.S. Department
of Energy. M.L. was supported by the Next Generation Ecosystem Experiments-Tropics, funded by the U.S. Department of Energy, Office
of Science, Office of Biological and Environmental Research. The U.S. Government retains, and the publisher, by accepting the article
for publication, acknowledges, that the U.S. Government retains a non-exclusive, paid-up, irrevocable, world-wide license to publish or
reproduce the published form of this manuscript, or allow others to do so, for U.S. Government purposes. Peter Thornton's participation was
supported as part of the Energy Exascale Earth System Model (E3SM) project, funded by the U.S. Department of Energy, Office of Science,
Office of Biological and Environmental Research Earth Systems Model Development Program area of Earth and Environmental System
Modeling. Michael Dietze acknowledges funding via the grant number NSF 2406258



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
