# Peer review of "Best practices in software development for robust and reproducible geoscientific models based on insights from the Global Carbon Project models"

_EGUsphere, 2025_

## Author Comment (AC2)

**Answers to Reviewer #2**

Reviewer comments are in black, our answers are in green, manuscript sections in *italic*, and manuscript additions in ***bold italic*** and removals in

In its own words, this article aims to combine the experience of land surface modelers and the expertise of professional software engineers in order to define the key principles and tools for improving software quality in the field of research. The authors conclude that it is possible to improve land surface models in areas such as automated testing, documentation, and reproducible workflows, but that inspiration can already be found in individual models, particularly the LPJ-GUESS model, which the authors consider the most successful in this field. This article is well written (with the help of ChatGPT, as explained in the acknowledgements – a honest and fair statement) and can certainly serve as a useful reference, but it can still be greatly improved. I actually tend to challenge the fact that "All [32] authors contributed in writing and editing the manuscript", or at least that many of them did it more than ChatGPT. This statement may seem provocative, but it stems from a few surprises:

We would like to thank the reviewer for their thorough assessment of our manuscript, which pointed out inconsistencies and flaws. We addressed these aspects and believe this significantly improved our manuscript.

We would like to respond to the four major criticisms, namely 1) the use of LLMs for text editing, 2) the usage of one specific model for the example, 3) the criticism of the capability of scientists in terms of programming, and 4) the situation of funding.
The comments by the reviewer were quite critical but raised important points, on most of which the authors agree. We are confident that addressing them has improved the paper by quite a lot. In the following we will address the comments in more detail:

First, we have to refute the claim about the author's contributions and AI use. Indeed, all authors contributed intellectually to different parts of the manuscript. The fact that still inconsistencies and flaws were found shows the importance of an external and unbiased peer review. We did use ChatGPT for language editing, but we have hundreds of emails, 20+ survey responses, and intermediate paper versions from well over one year to challenge that this paper or even parts of it were AI-generated. In fact, this paper started as a short paper by the first author on coding tips for modeling communities. An initial submission was challenged by the editors of GMD because it was not detailed enough, and it was suggested to include insights from other model communities. This led to a constantly growing manuscript with the insights of now 32 people, and offers a scope much larger than that of recently published papers that focus on single aspects (like Docker, version control, code review, FAIR, etc., see References 1–5 below).

Second, the reviewer wonders why the model LPJ-GUESS was stated as "most successful in the field". While we neither believe nor say this, we did realize now that we used it for most examples, simply because the first author is most familiar with it. We now removed the mention of this particular model in almost all cases because while the example figures or code snippets are based on this model, they are relevant for all models. In addition to our existing disclaimer about examples not being endorsements/criticism in the

introduction, we added more claims like this, and more clearly indicate that for the pipeline example (previously "showcase") we selected one model for pragmatic reasons and reduced the mentioning of which model we used for this to a minimum. We also worked in more references to the appendix (examples of all the model communities) to make sure benefits of numerous models are highlighted.

Third, the reviewer raises concerns about our criticism of the work/skill of the community. We realize now that the tone we picked was not ideal and more negative than it was intended to be (notably, considering that the author team represents numerous models, we were criticizing our own work). We have thoroughly revised the manuscript to adopt a neutral tone.

Fourth, the reviewer disagrees with our claims about the lack of time and money in this field, because they have enough time and money. For us, this highlights the relevance of including authors working with multiple models. The situation described by the reviewer does not align with the experience raised by multiple co-authors of this paper as well as that of surveys which we cite, therefore we prefer to keep this notion in the paper but extended the claims to show that experiences may differ..

In the following, we address the detailed comments of the reviewer.

- Why would land surface modelers claim that their models are "the Global Carbon Project models" (title)? Have they not read Table 4 of Friedingstein et al. (2023) or other papers in the series that many of them coauthored? This table lists many more types of models in GCP's Global Carbon Budget (GCB).
  - This is true. The editors claimed that our title was too click-baity because it initially was only called "*Best practices in software development for robust and reproducible geoscientific models*". We chose this initial title because our suggestions are useful for all sorts of models. We then added the last part to address this editor comment and, seeing this reviewer statement, we agree that this is misleading. We now specified this better: "*Best practices in software development for robust and reproducible geoscientific models based on insights from the Global Carbon Project**'s dynamic vegetation** models*"
- Why would land surface modelers explain that all models in geoscience including their own ones represent "processes that cannot be solved analytically" (l. 2)?
  - This is a fair point. We changed this to: *Computational models play an increasingly vital role in scientific research, by numerically simulating **highly complex** processes .*
- Why would land surface modelers suggest (l. 2) that their models are only used to study global change?
  - We are not stating this. We state that these models are used for this, not that they are **only** used for this. We made this clearer by saying, *Such models are fundamental in geosciences . **For instance, they can** offer critical insights into the impacts of global change on the Earth system today and in the future.*
- Why would land surface modelers from the GCB ignore the fact that some of them contribute to GCP's Global Methane Budget (GMB) and to GCP's Global Nitrous

Oxide budget, and that other biogeochemical models contribute to the GMB as well
(l. 8-9, 46-51)?

- These models contribute to a variety of community efforts that we could all
  mention. Here we simply focused on the GCB. We added a statement to
  indicate that these models are used for many other things as well:
  *Some of these models also contribute to the Global Methane Project (Saunois
  et al., 2024), the Global Nitrous Oxide Budget (Tian et al., 2024), and other
  community efforts, but the focus of this paper remains gathering the insights
  of the LSMs of the GCP.*

- Why would the modelers who, arguably, wrote Section A 1.7, have the left five lines
  in German in their section if they have properly read the paper?
  - An accident, this was a comment that was made in the last round of internal
    review (unlike the other rounds this did not include all co-authors, but just the
    core team) and was simply overlooked. It was removed. These lines had
    nothing to do with A1.7, but were about how to order the examples within A1.
    Because it was always a point of concern for us to give all models their space
    and not rank them.

- Why would they have left the table (1) describing their model with so little useful
  information? Rough number of code lines, rough code age, rough developer number
  (cf. l. 84), number of dedicated scientific programmers (cf. l. 421-422), rough number
  of systems on which the code has been ported, main configuration (site scale,
  regional or global scale, coupling with an Atmospheric General Circulation Model,
  usage as a component of an Earth System Model), and whether they are used in an
  operational framework or not, would also be interesting to the reader in the paper
  context (cf. l. 423-424).
  - We will add information on lines of code, developer number, and number of
    full-time scientific programmers to the Table.

- Why would the proud representatives of the 20 land surface models choose to elect
  one of them as the ideal model (l. 20, l. 68-69, or Section 7 pompously called "A
  showcase") without much justification? For example, Section A1.7, with its
  comprehensive suite of tests run daily, strikes me as impressive (or is the paragraph
  fake?): it is necessary to discuss the reasons why this model workflow is inferior to
  the one that was chosen.
  - We never state that one of the models is ideal but we understand why it could
    come across like this and have now addressed this in our revisions (see
    below). Also it seems that the indication of using chatGPT for text editing
    leads the reviewer to the conclusion that some of our paragraphs are "fake",
    which is simply untrue.
  - While we do not find the term "showcase" pompous, we changed it to
    "pipeline example".
  - We never mention that the model LPJ-GUESS is better than any of the
    others. The reviewer claims we call this model "ideal", or "most successful",
    but we neither believe nor say this in the text. The "pipeline example" uses
    LPJ-GUESS but the key message lies within the workflow not within the
    model used for the demonstration. Essentially, this example (and other
    examples within the manuscript) could have been given with any of the 20
    models. Our choice to use the model most familiar to the lead author is simply

pragmatic. To prevent any misinterpretation we have made changes throughout the manuscript which are highlighted below:

- ○ We changed the wording of the pipeline example to only mention LPJ-GUESS once, ensuring the focus is on the surrounding setup using Docker, Github Actions, and Snakemake:
  - ■ *This includes data pre-processing, a full model run  **using** a Docker container, and simple post-processing and data analysis with Jupyter notebooks*
  - ■ *A Github Actions pipeline running automated code cleanup (linting), unit tests, compilation checks, as well as the entire Snakemake pipeline (including  model run, pre- and post-processing).*
  - ■ *Mölder et al. (2021) provide an extensive explanation and examples of how to use Snakemake in scientific workflows, while we provide an example  **for LSM workflows** (see Sect. 7 and Figure 5)*
- ○ We further removed the mention of LPJ-GUESS in the abstract
  - ■ *We conclude with an open-source example implementation of these principles  **demonstrating** portable and reproducible data flows, a continuous integration setup, and web-based visualizations.*
- ○ We also termed the specific code examples as model-agnostic, since although they are based on LPJ-GUESS, these examples are relevant for all models:
  - ■ *Example of a C++ unit test, adapted from  **one of the models***
  - ■ *Example of a merge request on a continuous integration server, here on Gitlab for  **one of the models***
  - ■ *The pipeline consists of downloading, cropping, and mapping data to enable a model run *
    - ● *In Fig. 5 we also renamed pipeline step "run_lpj_guess" to "run_model"*
  - ■ *We conclude with an example of a unit-tested, documented, version-controlled, portable, and reproducible **model** workflow *
- ○ To further clarify that we don't find LPJ-GUESS superior (as well as the other tools that we selected for the pipeline examples) we have added the following disclaimer to the text:
  - ■ *This example can be used by modeling communities as an example for their own workflows. It contains a simple, but typical pipeline of an LSM (in this case, LPJ-GUESS), including data preprocessing, post-processing of model outputs, and publicly hosted interactive plots with minimal effort. **While this example demonstrates the use of a specific model and software tools the intent is not to suggest that the chosen model or tools are superior to others. Pragmatic choices were made based on the familiarity of the lead authors. We simply want to demonstrate how a combination of tools can be used to develop unique, robust, and reproducible modeling***

> ***pipelines. Our setup can be used as a guide together with all other sections of our paper, to develop unique pipelines for all models and all supporting frameworks.***
>
> - The section A1.7 is not fake, it was written by Vladislav Bastrikov from the ORCHIDEE modelling group. Yes, this is impressive, as is the buildbot system used by ICON-LAND, which is why we dedicated distinct sections to these workflows. In line with the other reviewer comments, we have now included more references to these examples in the main text.

- In general, why would they have left their experience in software engineering described in such a superficial way, even suggesting that none of them are professional software engineers (l. 10) ? Exploring the references and URL given Table 1, I see that some of these models have an history of integrating developments from an heterogeneous ecosystem of contributors over several decades (cf. l. 84-85), that several models have been developed under the coordination of weather centers and that some of them have been components of Earth system models of Coupled Model Intercomparison Project (CMIP): some of their developers must be particularly good at software development or the models would not have survived the diversity of their contributors and of their computing environments, and would not have had such challenging applications like CMIP. Actually, the text suggests (l. 4 and 31) that only scientists develop these models. I do believe that some professional software engineers should be credited as well and that some of the scientists involved, formally trained or not (l. 31-32), are also remarkable software developers (see also l. 421-422). The fact that they generalize the dull description of their experience in software engineering to "all sorts of scientific modeling and software" (l. 62-63) may be seen as arrogant. Developers of Numerical Weather or Ocean Prediction, for instance, would be mere amateurs? Come on! They deserve better comments, and the LSM models as well.

  - After reading this comment we do realize our negative tone with respect to the people working within modeling communities. This was not our intent at all. We changed the wording in many places to change the tone and to highlight that in many cases we were already referring to "professional software developers" from both within and outside the community:
    - *Scientists  **often** lack formal training*
    - *… **potentially** also leading to code that does not adhere to **the** high**est** technical standards …*

| Old Text | Revision |
|---|---|
| *By combining the experience of modelers from the respective research groups with the expertise of professional software engineers, we bridge the gap between software* | *We combine the experience of modelers from the respective research groups with the expertise of software engineers from tech companies to outline …* |

| | |
|---|---|
| *development and scientific modeling to outline …* | |
| *(not in previous version)* | *While larger model communities with large funding have more options to hire professional software developers, or invest in the training of their scientists, this is usually not the case for smaller groups.* |
| *To aid model communities in improving their practices, we consolidate insights from software development professionals and model developers of the GCP-models…* | *To aid model communities in improving their practices, we consolidate insights from software and model developers of the GCP-models and tech companies…* |
| *the concepts are applicable for all sorts of scientific modeling and software* | *the concepts are applicable to all fields of scientific modeling and software* |
| *Indeed, climate models were found to require more efforts in becoming "more readable, maintainable, and portable" (Easterbrook, 2010)* | *This can of course take different forms and vary in severity, but even climate models, which often exhibit high software quality, have been shown to benefit from more efforts in becoming "more readable, maintainable, and portable" (Easterbrook, 2010).* |
| *Its adoption in the LSM community remains limited but some LSMs have been "containerized" with Docker…* | *Some LSMs have already been "containerized" with Docker…* |

- Why would land surface modelers who are familiar with MIPs explain that MIPs allow them to "*understand* the range of uncertainty" (l. 227 – what does it mean?), while also forgetting to mention that a main outcome of MIPs is debugging the weakest models? A deeper investigation of MIP usefulness is certainly not out of the scope of this paper (l. 231).
  - We have adjusted the phrases: *Second, model intercomparison projects allow us to  quantify the range of uncertainty across models, indicating where models diverge the most,  possibly pinpointing to*

> *model-specific issues. Thus, they show* where model communities can learn from one another and what processes require most attention.
>   ○ We still believe a deeper investigation of MIP usefulness is the topic of another paper as it would encompass more topics than the topics presented here (software quality, maintainability, documentation, reproducibility, etc.) and as our paper is already 27 pages long.

I really urge the 32 coauthors to deepen their analysis for the benefit of the readers. In reality, the LSM authors could have presented their software work in a much more positive light and highlighted the merit and strength of their efforts, whereas the article reads like a simple request for funding. Note that in their work, LSM developers also face the bugs of commercial software, like compilers: nobody's safe.

We are confident that after our revisions, we now present our work in a more positive way and that our tips will be helpful for readers.

Additional detailed comments:

1. It would be worth explaining that the entry ticket for LSMs in the GCB is rather cheap, as explained in Section S.4.2 of Friedlingstein et al. (2023): "We apply three criteria for minimum DGVMs realism by including only those DGVMs with (1) steady state after spin up, (2) global net land flux (SLAND – ELUC) that is an atmosphere-to-land carbon flux over the 1990s ranging between -0.3 and 2.3 GtC yr-1, within 90% confidence of constraints by global atmospheric and oceanic observations (Keeling and Manning, 2014; Wanninkhof et al., 2013), and (3) global ELUC that is a carbon source to the atmosphere over the 1990s, as already mentioned in Supplement S.2.2. All DGVMs meet these three criteria." As a consequence, the quality of the selected LSMs is likely heterogeneous and their engineering support as well.
   ○ Thank you for this very important remark! We added this information, however not using the term "quality": *Notably, the entry requirements for a model to participate in the GCB are not overly rigorous (Friedlingstein et al., 2023). As a consequence, the participating LSMs span a wide range of model sizes and structural complexity, with differing levels of detail, engineering support, and team sizes.*
2. Surprisingly, the issue of competition with the private sector for the recruitment of skilled computer engineers is not addressed at all.
   ○ Good catch. A previous version of the manuscript did include this. We added this again in the introduction: *Furthermore, academia has to compete for skilled engineers with the private sector which can usually offer higher financial compensation (Merow et al., 2023).*
3. The challenge of rewriting hundreds of thousands of lines of code and the dilemma of scientists losing their understanding of the rewritten versions must be addressed.
   ○ Fair point. We now mention this in the "Developer documentation" section: *Good documentation is also crucial for ongoing developers to stay oriented in large, actively evolving codebases that may consist of thousands to hundreds of thousands of lines of code, maintained and extended by large groups of developers.*
4. Line 2: why is the discussion restricted to the impact of global change?

- ○ It isn't, as mentioned in an answer to your comments above, we simply provide examples what these models are used for, now expanded to other examples (see above)

5. Line 49: what are the critical insights offered by the GCB for policy-making?
   - ○ The detailed analyses and the timeliness. We changed this sentence to be more accurate: *Its primary activity is the annual Global Carbon Budget, which offers detailed analyses of the carbon cycle . **It informs the assessment reports of the Intergovernmental Panel on Climate Change (e.g., IPCC, 2023) and offers low-lag analyses of carbon emissions and uptake of large economic regions.***

6. Line 50: where is the land flux to the ocean computed by LSMs used in the GCB?
   - ○ We now removed this part. Some DGVMs may be able to compute relevant variables for this but it is not part of the GCP, as stated in their supplements: "Representation of the anthropogenic perturbation of [land ocean aquatic continuum] CO2 fluxes is however not included in the GOBMs and DGVMs used in our global carbon budget analysis presented here"

7. Line 33: the statement about the lack of recognition for software development in academia is unfair, in particular if one relates it to l. 255-256 about dedicated high-profile scientific journals for software development. There are also many calls for proposal in some countries or groups of countries for funding software developments or just to offer professional software support.
   - ○ see our answer to comment 9 below

8. Line 86 and 631-632: the lack of funding for positions dedicated to scientific programming is also a choice made by scientists who set priorities within their host organizations.
   - ○ In our experience, scientific model development fundings are much more frequent. This will in the long run lead to a discrepancy between the amount of people actively working on the modeled processes and the amount of people being in charge of maintaining a model's technical quality. Please see our answer to comment 9.

9. Line 91: around me, the time spent on software development or on software support calls by scientists is a choice. It may not be sufficient, but authors should not blame the system in the first place, but rather themselves.
   - ○ We are glad to hear that the referee's modeling group has flexibility on time spent on software development and support, however our experience shows that this may not be the case for every group.This only reinforces the relevance of including many modeling group team members as co-authors, as this issue was raised multiple times. To address this heterogeneity in funding landscape, we revised the text as follows:

     *Additionally, funding security varies considerably across modeling groups, depending on the country of the lead institution, funding agency priorities, whether or not the model is part of long-term research programs. Especially for modeling groups that lack continuous and steady funding, hiring and retaining dedicated scientific programming positions is challenging, and high turnover and insufficient staff may hinder model development and governance.*

10. Line 124-125: to detect where something is wrong, reference outputs need to be available, which is not always the case. Think about the adjoint code of a LSM routine for instance (except if one builds a heavy adjoint testing machinery around each routine).
    ○ This is true. In addition to our existing statement in section 3.3: "*depending on [...] availability of validation data*", we added a short paragraph on this matter: *To be clear, validation efforts critically depend on the availability and quality of reference data. It must also be recognized that commonly used reference datasets often involve their own modeling steps. For example, MODIS evapotranspiration is derived from a complex model based on observed leaf area index (Mu et al., 2013), FLUXNET gross primary productivity data relies on modeled partitioning of measured fluxes into gross primary productivity and respiration (Pastorello et al., 2020), and vegetation carbon datasets are usually derived from upscaling inventory data based on machine learning methods (e.g. Pucher et al., 2022). Thus, exact agreement with reference datasets is neither expected nor necessarily desirable. This paper cannot offer definitive guidance for this issue. It remains an active area of research that requires continued collaboration across communities working in modeling and those working with experiments and Earth observation.*
11. Line 219: netCDF has already been used above without any expansion of the acronym.
    ○ Good catch. We replaced the full expansion of the acronym since we believe that the name netCDF is a common file format that needs no further description.
12. Line 224: MIP has not been defined.
    ○ We changed this to the non-abbreviated term "model intercomparison project"
13. Line 455: missing punctuation mark.
    ○ Added punctuation mark.
14. Appendix A: the authors should try to better exploit/integrate this appendix within the main text.
    ○ We now include more detailed mentions of the examples from the appendix within the main text.
15. 6: this subsection seems to be out of scope. It should be either removed or rewritten in order to fit the paper.
    ○ We agree that the transition to this topic was not ideal. We added a better transition: ~~In recent years, science has been described as facing a reproducibility crisis affecting nearly all scientific disciplines (Baker, 2016). While the gravity of the problem, or at least the terminology, is debatable (Fanelli, 2018), scientific experiments are often becoming increasingly hard to reproduce. Being pieces of software, model runs and their analyses should be easily reproducible, but this is not always the case.~~
    *Geo-scientific models are usually run as parts of larger workflows. First, data from various sources need to be collected and pre-processed before the model can be run. Second, the models are usually run for sets of experiments, spanning for instance different input data (e.g., multiple climate change scenarios from various climate models) and model setups (e.g., different parameters). Finally, output data of the models needs to be post-processed, statistically analyzed, plotted, and*

> *so forth. This makes it hard to reproduce such model results, although theoretically they should be reproducible on a different machine. This section addresses this issue of reproducibility, which is not only relevant in geoscience, and sometimes even called a ``reproducibility crisis'' (Baker, 2016, but see Fanelli (2018)).*

16. In Table 1, the citation Vuichard et al has been duplicated and Hoffman et al (l. 213) has no year associated.
    ○ We corrected these mistakes.

We again thank the reviewer for their very thorough review of our manuscript, and hope our adjustments of the manuscript addresses all comments sufficiently.

**References:**

1. "make models public": https://pnas.org/doi/full/10.1073/pnas.2202112119
2. "how to use docker": https://dl.acm.org/doi/10.1145/2723872.2723882
3. "how to use version control": www.tandfonline.com/doi/full/10.1080/00031305.2017.1399928
4. "code review is useful": www.doi.org/10.1111/jeb.14230
5. "10 short tips for coding in science": https://dx.plos.org/10.1371/journal.pcbi.1006561